# Table as a Modality for Large Language Models

**Liyao Li[1]\*, Chao Ye[1]\*, Wentao Ye[1], Yifei Sun[1], Zhe Jiang[3], Haobo Wang[1]**
**Jiaming Tian[1], Yiming Zhang[1], Ningtao Wang[2], Xing Fu[2], Gang Chen[1], Junbo Zhao[1]†**
[1]Zhejiang University    [2]Ant Group    [3]University of Michigan
{liliyao, ye.chao, j.zhao}@zju.edu.cn

## Abstract

To migrate the remarkable successes of Large Language Models (LLMs), the community has made numerous efforts to generalize them to the table reasoning tasks for the widely deployed tabular data. Despite that, in this work, by showing a probing experiment on our proposed StructQA benchmark, we postulate that even the most advanced LLMs (such as GPTs) may still fall short of coping with tabular data. More specifically, the current scheme often simply relies on serializing the tabular data, together with the meta information, then inputting them through the LLMs. We argue that the loss of structural information is the root of this shortcoming. In this work, we further propose TAMO, which bears an ideology to treat the **ta**bles **a**s **a**n independent **mo**dality integrated with the text tokens. The resulting model in TAMO is a multimodal framework consisting of a hypergraph neural network as the global table encoder seamlessly integrated with the mainstream LLM. Empirical results on various benchmarking datasets, including HiTab, WikiTQ, WikiSQL, FeTaQA, and StructQA, have demonstrated significant improvements on generalization with an average relative gain of **42.65%**.

## 1 Introduction

Table reasoning, the process of generating task-specific responses based on one or more pre-structured tables text, has emerged as a key research area. This encompasses various tasks such as table question answering [Pasupat and Liang, 2015], table fact verification [Chen et al., 2019], text-to-SQL [Yu et al., 2018], and predictive tasks [Ye et al., 2024a, Li et al., 2022, Van Breugel and Van Der Schaar, 2024]. Classical methods employ baselines such as BART [Lewis et al., 2020] or T5 [Raffel et al., 2020] to generate answers, often augmented with external retrieval frameworks [Patnaik et al., 2024]. With the advent of large language models (LLMs), like GPT-4 [OpenAI, 2023] and LLaMA [Touvron et al., 2023], this field undergoes substantial transformations. These LLMs generally adopt identical strategies, which involve serializing tables into text formats, often using markdown-like markup languages to represent tables, occasionally accompanied by a few examples[Herzig et al., 2020], as shown in Figure 1.

However, through a principled and empirical observation, we find that existing *serialization strategies may cause LLMs to lose their understanding of structural semantics* of tables. This observation is conducted on a specially designed diagnostic benchmark, named *StructQA*. This benchmark considers a structural semantic property unique to tabular data: *permutation invariance*, which means that most tables should retain consistent semantics regardless of row and column reordering. Ideally, LLMs should maintain approximately the same downstream task performance for structurally equivalent tables. However, as illustrated in Figure 2, our experiments reveal: leading LLMs, including Llama2-7B [Touvron et al., 2023], GPT-3.5 [OpenAI, 2022], GPT-4, and even TableLlama [Zhang et al.,

---

\*Equal contribution.
†Corresponding author.

2023b]—trained specifically for table tasks-demonstrate significant performance degradation when presented with permuted versions of the same table. While resisting these perturbations is trivial for humans, these models excluding GPT-4, show answer robustness below **40%**. This phenomenon indicates the ***significant limitations in LLMs' generalization capability for table comprehension through serialization***.

This limitation stems from a fundamental mismatch: ***tables are inherently structured data with permutation invariance***, *while text serialization cannot naturally preserve this property*. Although this concept has been recognized in previous research [Herzig et al., 2020, Yang et al., 2022], it has been largely overlooked in contemporary LLM development. When faced with structural perturbations, LLMs exhibit hallucinations [Huang et al., 2023] and unstable reasoning patterns, indicating their limited generalization of tabular structure.

To tackle this issue, we propose a novel perspective: ***encode tables as an independent modality to integrate their complex relational structures***. Just like images and audio which contain rich semantic information, tables possess inherent structural nuances that textual serialization fails to represent alone. The multimodal large language models (MLLM) learn the semantics of specialized modalities through separate encoding architectures and align different modalities in a unified and more expressive embedding space. By employing a similar approach, we can bridge the gap in LLMs' comprehension and achieve a holistic understanding of tables' structure comparable to human cognition through learnable table features.

In this paper, we propose TAMO[3], a pioneering tabular language model framework to reimagine **Ta**ble representation **a**s **a**n independent **Mo**dality. TAMO leverages theoretically permutation-invariant hypergraph structures to independently capture the intricate relationships and global structures within tabular data. Further, we integrate this hypergraph-based encoding into LLMs through learnable features, achieving dynamic and efficient injection of structural information without tuning the LLM's fixed parameters. We exhibit extensive empirical validation on five table reasoning datasets[4]. TAMO demonstrates substantial performance improvements against previous baselines—up to a **42.65% increase** in average performance. Meanwhile, our methodology validates superior efficacy and broad applicability when integrating hypergraph-encoded tables with diverse LLMs.

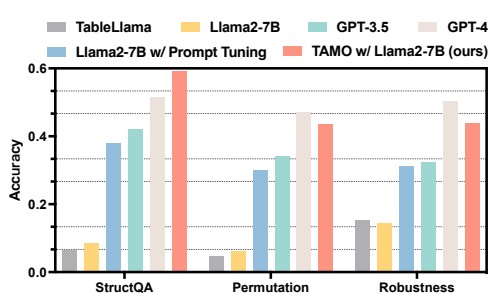

Figure 1: Current tabular LLMs oversimplifies tables into text sequences, ignoring structured information and hindering basic table cell localization tasks. This work is the first to direct table structure integration into LLMs.

Figure 2: We evaluated LLMs' understanding of table structures through our *StructQA* dataset (Section 3.1). The evaluation focused on permutation invariance by assessing answer robustness across randomly permuted rows and columns. Results show that TAMO achieves state-of-the-art performance, comparable with GPT-4.

**Contributions.** ***Benchmark***: We introduce StructQA, the first open-source benchmark on the robust tabular structure understanding. Our findings reveal that current LLMs struggle with this human-friendly task. ***Position***: Our research represents a pioneering step in integrating tables as an independent modality into LLMs. ***Methodology***: We explore the semantic alignment of tabular structures in LLMs' embedding space via

---

[3]Code and datasets are on `https://github.com/liyaooi/TAMO`.

[4]Hitab [Cheng et al., 2022], WikiTQ [Pasupat and Liang, 2015], WikiSQL [Zhong et al., 2017], and FeTaQA [Nan et al., 2022] and our proposed *StructQA* benchmark (Section 3.1).

hypergraph architectures, effectively modeling complex relational patterns across diverse table tasks. *Feasibility*: Experiments show that TAMO enhances LLMs' generalization on table reasoning by encoding structure-invariant table representations. We demonstrate the *generality* of our framework in two primary aspects: (1) **Robust Generalization** across structural variations, such as the permutations tested in our proposed StructQA benchmark, and (2) **Architectural Generality**, functioning as a "plug-and-play" module that integrates with diverse decoder-only LLMs without intrusive architectural modifications.

## 2 Methodology

### 2.1 Problem Definition

Following Wang et al. [2024b], table reasoning can be defined as a unified task that acts on samples formatted as triplets $(\mathcal{T}, \mathcal{Q}, \mathcal{A})$. Here, $\mathcal{T}$ represents a pre-structured table containing information clearly organized in rows and columns, with cell types encompassing numerical values, text entries, and dates. $\mathcal{Q} = \{q_1, q_2, ..., q_m\}$ denotes the question or statement related to the table $\mathcal{T}$, typically in a natural language sequence with $m$ tokens. Meanwhile, $\mathcal{A}$ is the expected answer or output of $\mathcal{Q}$, usually simplified into an $n$-tokens sequence $\{a_1, a_2, ..., a_n\}$. Briefly, given the table $\mathcal{T}$ and the question $\mathcal{Q}$, the objective of table reasoning is to predict the corresponding answer $\mathcal{A}$, i.e., $p(\mathcal{A}|\mathcal{T}, \mathcal{Q})$.

### 2.2 Hypergraph-enhanced Table Encoder

A table encoder is essential for our multimodal tabular LLMs paradigm. To develop the table encoder capable of learning structural information, we first address a fundamental question: "*How to define the structural properties in tabular data?*" As illustrated in Figure 3, we provide the answer based on prior human observations: (i)-most real-world tabular data possess a *hierarchical structure*, with ordinary flat tables being a special case of this hierarchy; (ii)-cells within each hierarchy and hierarchies at the same level exhibit *permutation invariance*. For example, arbitrarily swapping rows or columns in a table does not distort its original meaning. This implies that learning the relationships between table cells should

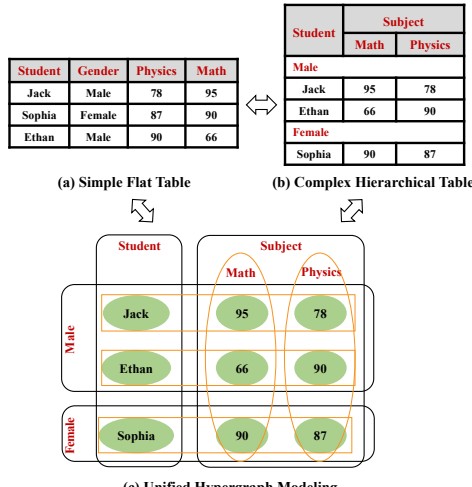

**(a) Simple Flat Table**    **(b) Complex Hierarchical Table**

**(c) Unified Hypergraph Modeling**

Figure 3: An example of converting arbitrary simple or complex tables into hypergraphs. A simple flat table is a special case of the complex hierarchical table. A hyperedge (e.g., table headers) in the hypergraph is a set of regular nodes. We construct the corresponding hypergraph format according to the hierarchical relationships of the table.

not be pairwise but rather set-based. Building on the inherent hierarchical structure of tables, we introduce the **hypergraph** [Yadati et al., 2019] architecture to model tabular data. This approach incorporates both *high-order hierarchical structure* and *permutation invariance* as inductive biases, enabling the precise modeling of complex structural properties in tabular data. For the first time, it allows us to successfully model all types of tables, from simple flat tables to complex hierarchical forms [Cheng et al., 2022].

We re-construct the structure of tabular data via hypergraph. Specifically, a hypergraph $\mathcal{G} = (\mathcal{V}, \mathcal{E})$ consists of a set of nodes $\mathcal{V}$ and hyperedges $\mathcal{E}$. Each hyperedge $e \in \mathcal{E}$ is a subset of $\mathcal{V}$, i.e., $e \subseteq \mathcal{V}$. For a table $\mathcal{T}$, we represent each leaf cell, defined as a cell that does not contain any other cells within the hierarchy, as a node $v \in \mathcal{V}$ and each branch cell, defined as a cell that contains other cells within the hierarchy, as a hyperedge $e \in \mathcal{E}$. Each hyperedge $e$ consists of nodes that belong to its hierarchical level. For example, in a simple flat table, each table cell is a node, and each column or row is a hyperedge encompassing all nodes within that column or row. Under this modeling, altering rows or columns maintains a consistent graph structure (both nodes and edges), effectively reflecting the *permutation invariance* of tables.

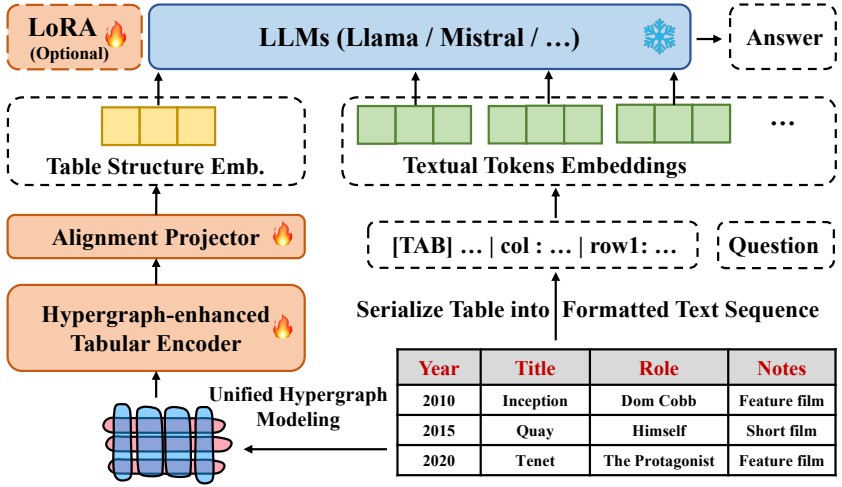

Figure 4: The proposed framework for tabular LLMs, TAMO. Given a table input, the hypergraph-enhanced table encoder (Section 2.2) is used to capture the unique structure properties of the table modality. Simultaneously, we serialize the original table into a formatted text sequence. Finally, we input both the table structure and textual embeddings into LLMs, generating answers using the next token prediction paradigm. LoRA is optional.

Furthermore, to model information propagation within the hypergraph structure, we employ Hyper-Trans [Chen et al., 2024], a hypergraph-structure-aware transformer, as the backbone architecture of the hypergraph-enhanced table encoder. The encoder architecture integrates two distinct multiset functions [Chien et al., 2021] to effectively capture *higher-order hierarchical structures* in hypergraphs. These multiset functions, characterized by their *permutation invariance* property, are combined sequentially as depicted in Eqa.1 and Eqa.2. Each layer of the table encoder comprises two primary components, with the initial component implementing a multiset function that aggregates node-level information to update hyperedge representations:

$$\mathbf{x}_e^{t+1} = Fusion(\mathbf{x}_e^t, Multiset_1(\{\mathbf{x}_v^t \mid v \in e\})), \tag{1}$$

where $t$ refers to the current layer number; $\mathbf{x}_v$ is the embedding of the node $v$; $\mathbf{x}_e$ is the embedding of the hyperedge $e$; the *fusion* layer is employed to integrate hyperedge information from the last layers, typically utilizing a multilayer perceptron (MLP) network.

The second part is another multiset function that aggregates hyperedge information to update node representations:

$$\mathbf{x}_v^{t+1} = Multiset_2(\{\mathbf{x}_e^{t+1} \mid v \in e\}). \tag{2}$$

Finally, we use the Set Transformer [Lee et al., 2019] to parameterize these multiset functions for learning. Each set attention block is defined as:

$$Multiset(\mathbf{X}) = LayerNorm(\mathbf{H} + rFF(\mathbf{H})),$$
$$H = LayerNorm(\mathbf{X} + MultiHead(\mathbf{S}, \mathbf{X}, \mathbf{X})), \tag{3}$$

where $\mathbf{S}$ is a trainable parameter vector; $rFF$ is the row-wise feedforward layer; $LayerNorm$ is layer normalization [Ba et al., 2016]; $MultiHead$ is the multi-head attention mechanism [Vaswani et al., 2017]. By facilitating the mutual propagation of information between nodes and hyperedges, the model effectively learns the complex hierarchical relationships among table cells thus outputting learnable table features. This approach conceptually reframes the table from a serialized string into a distinct modality, which we position as a complete multimodal reasoning framework distinct from modality-specific encoders. A detailed discussion on this positioning is provided in Appendix D.1.

## 2.3 A Modality Interface for Integrating Table Structure Representations with LLMs

Most LLMs [Meta, 2024, Jiang et al., 2023a, OpenAI, 2022, 2023] are pre-trained on large-scale unlabeled corpora in an *autoregressive* manner, thereby learning rich linguistic structures and patterns. To maximize the utilization of LLMs' powerful text understanding and reasoning capabilities for table reasoning tasks, we design a fully *autoregressive* interface to integrate structure representations from the tabular modality with LLMs for table reasoning tasks. The overall framework of our proposed TAMO is shown in Figure 4. We inject the structure representations learned by the hypergraph-enhanced table encoder in Section 2.2 into the LLMs in a manner similar to the soft prompt [Lester et al., 2021]. *This allows the LLMs to globally perceive the structural information of the tabular data before reading the textual information*, thereby enhancing their understanding and reasoning abilities regarding tabular tasks.

**Aligning Table Structure Representations to LLM Semantic Space.** Assuming the node representations obtained through the table encoder are $\hat{\mathbf{X}}_{\mathcal{V}} = \{\hat{\mathbf{x}}_v | v \in \mathcal{V}\} \in \mathbb{R}^{|\mathcal{V}| \times d_g}$, and the hyperedge representations are $\hat{\mathbf{X}}_{\mathcal{E}} = \{\hat{\mathbf{x}}_e | e \in \mathcal{E}\} \in \mathbb{R}^{|\mathcal{E}| \times d_g}$. $d_g$ is the hidden dimension of the table encoder. We use a multilayer perceptron (MLP) network to learn the transformation of table structure representations $\mathbf{X}_{st}$ into the semantic space, where we adopt a single-layer MLP as the alignment projector for simplicity and efficiency:

$$\mathbf{X}_{st} = MLP(Pooling(\hat{\mathbf{X}}_{\mathcal{V}}, \hat{\mathbf{X}}_{\mathcal{E}})) \in \mathbb{R}^{d_l}, \tag{4}$$

where *pooling* is an information aggregation function for nodes and hyperedges, set up as *mean pooling* in our experiment; $d_l$ is the hidden dimension of LLMs.

**Generating Answers based on both Tabular and Textual Modality Information.** Following previous works [Zhang et al., 2023b, Wang et al., 2024b, Herzig et al., 2020], we serialize tabular data into formatted text sequences and obtain the text embeddings of tabular data $\mathbf{X}_{tt} \in \mathbb{R}^{L_s \times d_l}$ through the LLMs' embedding layer. $L_s$ is the length of text sequences. For questions in natural language form, we obtain the corresponding question tokens $\mathbf{X}_{qt} \in \mathbb{R}^{L_q \times d_l}$ through the embedding layer similarly. $L_q$ is the length of question sequences. The final answer is generated following the next token prediction paradigm:

$$p(\mathcal{A}|\mathcal{T}, \mathcal{Q}) = \prod_i^n p(a_i \mid \mathbf{X}_{st}, \mathbf{X}_{tt}, \mathbf{X}_{qt}, a_{j<i}), \tag{5}$$

where $n$ is the number of answer tokens $\mathcal{A} = \{a_1, a_2, ..., a_n\}$. During training on downstream table reasoning datasets, we can choose to freeze the parameters of the LLMs and only learn the table encoder and alignment layers. We retain both input streams as they serve complementary, non-redundant roles. The serialized text $\mathbf{X}_{tt}$ provides the fine-grained semantic content (the "what"), which LLMs are exceptionally skilled at processing. The hypergraph-based structure embedding $\mathbf{X}_{st}$ provides the global relational context (the "where"), such as row-column relationships and hierarchies, which is inherently lost during serialization. The necessity of both modalities is empirically validated in our ablation study (Appendix C.4), which shows that the graph-only approach fails on generative tasks and the full TAMO model significantly outperforms the text-only counterpart, confirming the synergistic benefit. *This method allows us to capture structure representations in the tabular modality while integrating them with LLMs in a **cost-effective** and **scalable** manner.*

## 3 Experiments

In this section, we will demonstrate the advantages of treating tables as an independent modality. Section 3.1 introduces our novel benchmark, StructQA, designed to evaluate LLMs' understanding of table structures and their robustness. Sections 3.3 presents the performance gains of our approach across mainstream datasets and fine-tuning methods. Section 3.4 explores the interpretability of our method through attention visualization. Section 3.5 showcases the robust performance of our method under different fine-tuning techniques. Section 3.6 discusses the ability of the hypergraph-enhanced table encoder to extract table structure information.

## 3.1 StructQA: Table Structure Understanding

We present ***StructQA***, a novel benchmark focusing on table structure understanding, containing 7500 QA pairs from 500 tables across 5 structural reasoning tasks as described in Table 1. Unlike prior table QA datasets [Pasupat and Liang, 2015, Cheng et al., 2022], *StructQA* evaluates models' capabilities through three dimensions: **direct performance**, **permutation** accuracy under row/column shuffling, and answer **robustness** before and after permutation. This benchmark also addresses potential data contamination concerns [Ye et al., 2024b] present in existing datasets.

Figure 2 indicates the performance of mainstream LLMs on *StructQA*. We find that current text-based method do not perform well on this dataset, with overall answer robustness below **40%**. This phenomenon may suggest that current approaches face significant limitations in understanding tabular data.

## 3.2 Experimental Setup

To enhance the model's comprehension of structured data, we propose TAMO (refer to Section 2.2 for details). This section primarily outlines the experimental setup for the experiments of TAMO.

**Datasets & Metrics.** To conduct a more comprehensive assessment, we choose five datasets[5], including ***StructQA***, ***HiTab***, ***WikiTableQuestions*** (WikiTQ), ***WikiSQL***, ***FeTaQA***, for our experiments. To establish the performance upper bounds of TAMO for different tasks independently, we trained separate instances from scratch on each task-specific training set and evaluated them on the corresponding test sets[6].

(1) ***Cell location***: identify cell value by row number and column name.
(2) ***Column lookup***: identify the column based on row number and cell value.
(3) ***Row lookup***: identify the row based on the column name and cell value.
(4) ***Column comprehension***: summarize all distinct values in a column based on the column name.
(5) ***Row comprehension***: summarize all distinct values in a row based on the row number.

Table 1: Five different types of structure reasoning tasks in the *StructQA* dataset. More details are in Appendix A.

**Competing Methods.** To demonstrate that incorporating tabular modality into LLMs, referred to as *tabular language models*, can enhance performance in table reasoning tasks, we compare **TAMO** against using only pure text modality in four different settings: (i) **Inference Only (Zero-shot)**: directly reasoning on serialized table sequences and questions; serves as a diagnostic baseline (primary claims rely on stronger tuned comparisons). (ii)-***Frozen LLM***: comparing with prompt tuning [Lester et al., 2021], which adds some parameterized and trained tokens in front of serialized table sequences. (iii)-***Tuned LLM (LoRA)***: using LoRA [Hu et al., 2021] to finetune the parameters of LLMs. We add optional LoRA in our method as $\text{TAMO}^+_{LoRA}$. (iv)-***Tuned LLM (SFT)***: supervised finetuning of all parameters of LLMs. $\text{TAMO}^+_{SFT}$ means supervised training of TAMO and LLMs jointly.

A key baseline in our comparison is TableLlama [Zhang et al., 2023b], which represents the current state-of-the-art (SOTA) across multiple table reasoning tasks through supervised fine-tuning (SFT) on extensive tabular datasets. For a fair comparison with TableLlama, we implement TAMO using the same base model, Llama2-7B. We provide additional performance analysis across more advanced LLMs in Appendix C.8. We further evaluate our approach against powerful general-purpose LLMs like GPT-3.5-turbo-0125, GPT-4-turbo-2024-04-09, GPT-4.1 and DeepSeek-R1 [Guo et al., 2025]. Regarding the "Specialist SOTA" results presented in Table 2, these refer to state-of-the-art methods specifically tailored for each dataset: TableLlama [Zhang et al., 2023b] for HiTab, CABINET [Patnaik et al., 2024] for WikiTQ and FetaQA, and SeaD [Xu et al., 2022] for WikiSQL. We included these to highlight different methodological directions, contrasting with TAMO's focus on enhancing the LLM's end-to-end understanding via modality integration. Unlike TAMO's approach, these specialist methods often rely on external modules (like scorers or specific pre-processing steps) or task-specific objectives, which limit their direct applicability across the full range of table reasoning tasks without adaptation.

---

[5]Details of these datasets can be found in Appendix B.
[6]Appendix C.6 demonstrates that TAMO exhibits notable cross-dataset generalization capabilities.

| Setting | Dataset Task Type Evaluation Metric | StructQA Structural QA Accuracy | HiTab Hierarchical QA Accuracy | WikiTQ Table QA Accuracy | WikiSQL Table QA Accuracy | FetaQA Free-form QA BLEU |
|---|---|---|---|---|---|---|
| Inference Only | Zero-shot | 8.60 | 7.77 | 14.50 | 21.44 | 20.08 |
| Frozen LLM | Prompt tuning | 37.80 | 26.26 | 29.86 | 61.24 | 29.94 |
| | **TAMO** | 59.07 | 48.86 | 37.06 | 76.45 | 36.52 |
| | $\triangle_{Prompt\ tuning}$ | ↑ 56.27% | ↑ 86.06% | ↑ 24.11% | ↑ 24.84% | ↑ 21.98% |
| Tuned LLM (LoRA) | LoRA | 45.67 | 50.76 | 37.13 | 57.10 | 35.80 |
| | **TAMO**$^{+}_{LoRA}$ | 70.80 | 59.22 | 43.53 | 84.43 | 37.43 |
| | $\triangle_{LoRA}$ | ↑ 55.03% | ↑ 16.67% | ↑ 17.24% | ↑ 47.86% | ↑ 4.55% |
| Tuned LLM (SFT) | TableLlama[2023b] | 6.47 | 63.76 | 31.22 | 46.26 | 38.12 |
| | SFT | 62.73 | 54.80 | 43.28 | 79.86 | 37.37 |
| | **TAMO**$^{+}_{SFT}$ | **71.60** | **63.89** | **45.81** | **85.90** | **39.01** |
| | $\triangle_{SFT}$ | ↑ 14.14% | ↑ 16.59% | ↑ 5.85% | ↑ 7.56% | ↑ 4.39% |
| Others | GPT-3.5 | 41.93 | 43.62* | 53.13* | 41.91* | 26.49* |
| | GPT-4 | 51.40 | 48.40* | 68.40* | 47.60* | 21.70* |
| | GPT-4.1 | 60.33 | 60.54 | 68.14 | 71.21 | 36.75 |
| | DeepSeek-R1 | 57.47 | 63.89 | 75.76 | 71.91 | 13.10 |
| | Specialist SOTA | – | 64.71[2023b] | 69.10[2024] | 92.07[2022] | 40.50[2024] |

Table 2: Results on our table structure understanding dataset *StructQA* and four table reasoning benchmarks. TAMO adds additional table modality information compared to the pure text baseline. Specialist SOTA refers to methods that design models and training tasks specifically for each dataset. "*" indicates data sourced from Zhang et al. [2023b]. The first best result for each task is highlighted in **bold** and the second best result is highlighted with an underline.

## 3.3 Main Results

Main results are exhibited in Table 2, we conclude that:(i)- *Explicitly inputting the table modality significantly enhances LLM's performance in various table reasoning tasks*; (ii)-TAMO *significantly outperforms the SFT models that rely solely on the text modality*; (iii)-TAMO *is competitive with specialist SOTA methods, highlighting the utility of using hypergraphs to model complex table structure relationships*.

Across *all* datasets, TAMO achieves substantial improvements in *both* frozen and tuned LLM settings. For example, TAMO shows an average improvement of **+42.65%** over inputting pure text modality on the frozen LLM setting, with a maximum improvement of **+86.06%** on the HiTab dataset. In the tuned LLM setting, both TAMO$^{+}_{LoRA}$ and TAMO$^{+}_{SFT}$ show substantial improvements, outperforming the pure text modality by an average of +28.27% and +9.71%, respectively. Meanwhile, TAMO$^{+}_{SFT}$ achieves SOTA performance across all tasks under our settings and TAMO$^{+}_{LoRA}$ secures a close second on 3 out of 5 datasets. This reveals that **TAMO** *provides LLMs with more comprehensive table embeddings that remains unattainable through text sequences* regardless of training setting.

The Llama2-7B based TAMO$^{+}_{SFT}$ achieves closed SOTA performance on HiTab, FetaQA, and WikiSQL, where HiTab is a complex hierarchical table dataset. This indicates that hypergraph-enhanced table encoder can effectively learn complex hierarchical relationships within tables, thus further improving the model's accuracy in table reasoning tasks. Although slightly behind the specialist SOTA methods on the other datasets, it's worth noting that they all utilized *dataset-specific* model architectures, training methods, or other enhancement tricks. Additionally, TAMO$^{+}_{LoRA}$ and TAMO$^{+}_{SFT}$ consistently surpass GPT-3.5 and GPT-4 on 4 out of 5 datasets. For example, TAMO$^{+}_{SFT}$ achieves an average improvement of over **+19.83** score compared to GPT-3.5. When compared against the more recent powerful foundation models, GPT-4.1 and DeepSeek-R1, our 7B-parameter TAMO$^{+}_{SFT}$ demonstrates strong competitiveness, particularly on structure-centric benchmarks. As shown in Table 2, TAMO$^{+}_{SFT}$ significantly outperforms both GPT-4.1 (71.60 vs 60.33) and DeepSeek-R1 (71.60 vs 57.47) on StructQA, and notably surpasses them on WikiSQL (85.90 vs 71.21 and 71.91, respectively). While these larger models show stronger performance on knowledge-intensive tasks like WikiTQ (e.g., DeepSeek-R1 achieves 75.76), the competitive results achieved by our much smaller model underscore the effectiveness of the TAMO architecture in enhancing structural reasoning capabilities.

| Input : [ T AB ] col : Pick \| Player \| Position \| National ity \| NHL team \| College /j un ior / club team \| [ SEP ] \| 27 \| Rh ett War re ner \| Defence \| Canada \| Florida Panthers \| Sask atoon Blades ( W HL ) \| [ SEP ] … \| 35 \| Josef Mar ha \| Center \| Czech Republic \| Quebec Nord iques \| D uk la J ih lava ( C zech Republic ) \| [ SEP ] \| 36 \| Ryan Johnson \| Centre \| Canada \| Florida Panthers \| Thunder Bay Flyers ( US HL ) … Question : What are the national ities of the player picked from Thunder Bay Flyers ( ush 1 ] |
|:---|

**Inference only**

| [table_structure_token] … Input : [ T AB ] col : Pick \| Player \| Position \| National ity \| NHL team \| College /j un ior / club team \| [ SEP ] \| 27 \| Rh ett War re ner \| Defence \| Canada \| Florida Panthers \| Sask atoon Blades ( W HL ) \| [ SEP ] … \| 35 \| Josef Mar ha \| Center \| Czech Republic \| Quebec Nord iques \| D uk la J ih lava ( C zech Republic ) \| [ SEP ] \| 36 \| Ryan Johnson \| Centre \| Canada \| Florida Panthers \| Thunder Bay Flyers ( US HL ) … Question : What are the national ities of the player picked from Thunder Bay Flyers ( ush 1 ) |
|:---|

**TaMo (Ours)**

Figure 5: A real visualization case in the WikiSQL dataset results of attention weights from other input tokens to the label answer cell ''Canada''. Intuitively, the darker the color, the more closely the token is associated with ''Canada''. We observe that with the ''[table_structure_token]'' of TaMo, the LLM better focuses on information relevant to the correct answer, as indicated by the darker background colors associated with those tokens.

## 3.4 Case Study

To further investigate the table encoder's comprehension of tabular data, we conduct a visual analysis of specific cases. Specifically, we visualize the attention importance of all input tokens for the correct answer as perceived by the LLMs. During this process, we adopt the visualization method from the PromptBench [Zhu et al., 2023b], which uses the gradients of the input embeddings to estimate token importance. We randomly select a sample from the WikiSQL test sets for visualization analysis, where the base method (inference only) is incorrect but TaMo is correct.

As shown in Figure 5, we can find: (i)-TaMo thinks ''Canada'' (correct answer) and ''US HL'' (relevant contextual information) tokens are more important for the final answer, while the base method largely ignores these crucial tokens. (ii)-TaMo shows a certain level of attention to ''[table_structure_token]'', and adding ''[table_structure_token]'' affects the importance distribution of other input tokens, prompting LLMs to focus more on tokens relevant to the correct answer. We observed some error cases with the LoRA setting that resemble those shown above. For example, when the correct answer is far from the question in the serialized input, TaMo can utilize the overall table structure to locate the correct answer, compared to LoRA in text-only mode, which primarily focuses on the content immediately before and after the question. This case study indicates that the *structural information learned from* TaMo *enhances LLMs' reasoning capabilities by optimizing attention relationships for key table information*, thereby indicating potential for mitigating hallucination phenomena.

## 3.5 Generalization to Structural Variations

Compared to image/text data, *permutation invariance*—any permutation of the rows and columns does not change the original interpretation of the table—is a unique structural property of tabular data. To further explore whether TaMo can effectively perceive table structure information, we construct experiments to assess its robustness regarding permutation invariance. Specifically, we use the permutation version test set by randomly shuffling the rows and columns of tables in the StructQA test set (the training set is unchanged). In the frozen LLM setting, we compare the performance of TaMo with pure text modality methods (inference only & prompt tuning) on the new test set and check the consistency of answers after permutation. Results are shown in Figure 2 and Figure 6, we find that for *both* frozen LLMs and tuned LLMs (LoRA and SFT), TaMo consistently outperforms pure text modality methods. Additionally, TaMo demonstrates the best robustness in maintaining consistent results after permuta-

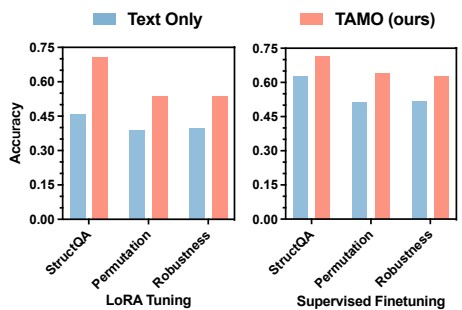

Figure 6: Evaluate the robustness of TaMo to permutation invariance on the StructQA dataset. *Permutation*: randomly permuting rows and columns in the StructQA test set. *Robustness*: the proportion of samples that remain consistent after random permutation.

tion. These indicate that **TAMO** *can enable superior generalization of LLMs across diverse table structural variations.*

## 3.6 Generalizable Table Encoding Analysis

To further validate the table encoder's effectiveness in learning table structure, we conduct a series of additional experiments. Specifically, we propose a binary classification task to predict whether cells belong to specific rows or columns. For this experiment, we use the WikiTQ [Pasupat and Liang, 2015] as base dataset, train all models for 50 epochs with a learning rate of 3e-4. Finally, we evaluate them with F1 metric.

The results are shown in Table 3. Without table representation learned from the encoder, a baseline MLP classifier only achieve **5.39%** F1 score, while a randomly initialized table encoder with MLP head reach **49.73%**. Most notably, when using our pretrained table encoders from various datasets (StructQA, HiTab, WikiTQ, WikiSQL, and FetaQA) with an MLP classifier, all models exceed 60% F1 score, with StructQA achieving the highest at **71.32%**. The superior performance on StructQA can be attributed to its focused structural learning with minimal reasoning complexity. These results, evaluated consistently on the WikiTQ test set, demonstrate that **TAMO***'s hypergraph-based table embeddings effectively encode and generalize structural relationships across different datasets.*

| Settings | F1 Score |
|---|---|
| MLP head | 5.39 |
| + randomly enc. | 49.73 |
| + pretrained enc. | |
| StructQA | **71.32** |
| HiTab | 66.39 |
| WikiTQ | 62.63 |
| WikiSQL | 68.00 |
| FetaQA | 64.99 |

Table 3: Evaluation of table structure representations learned from encoders pretrained on different datasets, measured on the WikiTQ test set.

Combined with the interpretability analysis in Section 3.4, these findings suggest that table structure representations can enhance LLMs' understanding of table and answer localization capabilities during reasoning, aligning with observations from previous work [Yang et al., 2022]. Additional ablation studies are provided in Appendix C[7][8].

## 4 Related Work

**LLM-based Table Reasoning.** Recent advances in Large Language Models (LLMs) have led to their widespread adoption in tabular reasoning tasks [Zhang et al., 2024], establishing Tabular Large Language Models as the dominant approach. These methods primarily follow two strategies: fine-tuning on tabular data and prompt engineering. The fine-tuning approach enhances LLMs' structured data capabilities through supervised training on tables [Zhang et al., 2023b, Zhuang et al., 2024, Wu and Feng, 2024, Sarkar and Lausen, 2023], exemplified by TableLlama's [Zhang et al., 2023b] generalist model trained on diverse real-world tables. Several recent parallel studies [Long et al., 2025, Jin et al., 2025, Xu et al., 2025, Majee et al., 2025, Huang et al., 2025] have independently explored modeling fine-grained table semantics (e.g., cell-level or column-level embeddings), typically in task-specific contexts such as text-to-SQL or table type classification. However, these methods generally rely on heavy pretraining or multi-stage training pipelines. In contrast, our method adopts a plug-and-play design that provides global tabular representations and does not require any pretraining. A detailed comparison is provided in Appendix D.2. The prompt engineering approach instead leverages carefully crafted prompts to enhance LLMs' reasoning over tables in specific scenarios [Ni et al., 2023, Wang et al., 2024b, Jiang et al., 2023b, Zhang et al., 2023b, Cheng et al., 2023]. Representative works include Dater [Ye et al., 2023], which decomposes tables into subtables, and Chain-of-Table [Wang et al., 2024b], which integrates chain-of-thought reasoning with programmatic methods.

**Table Encoder.** In recent years, numerous studies have explored effective methods for encoding and understanding tabular data. TaBert [Yin et al., 2020] adopts a dual-encoder framework that separately processes textual and structural elements of tables, improving table comprehension through masked language modeling. TabNet [Arik and Pfister, 2021] utilizes a novel iterative masking

---

[7]Appendix C.2 presents the relatively low computational overhead introduced by the TAMO framework.

[8]Appendix C.7 demonstrates the notable performance gain of TAMO in the multi-table QA setting.

attention mechanism to select important features. HyTrel [Chen et al., 2024] extends this concept by using hyperedges to capture richer interactions among simple flat table cells, resulting in enhanced representations for relational data. However, all these table encoders cannot handle joint text and table understanding tasks like table question answering. They are primarily used to encode raw tabular data into a low-dimensional vector space to get better table representation. As discussed in Section 1, existing approaches often serialize tables into text, losing structural information. To address this, we propose TAMO, a multimodal framework that integrates structural and textual semantics into LLMs. TAMO is compatible with various table encoders; in this work, we use HyTrel for its effectiveness. While above encoders focus on learning effective table representations, seminal works such as TAPAS [Herzig et al., 2020] (BERT-based) and TAPEX [Liu et al., 2022] (BART-based) successfully integrated structural awareness within the then-dominant encoder-only or encoder-decoder language model frameworks. Our work, TAMO, differs by proposing a modality interface specifically designed for integrating table structure into modern *decoder-only* LLMs. We provide empirical comparisons against TAPAS and TAPEX on our StructQA benchmark in Appendix C.5 to highlight the advancements and specific challenges addressed within this newer architectural paradigm.

## 5  Limitations

While TAMO enhances frozen-parameter LLMs' table reasoning capabilities via hypergraph encoders and learnable features, it has several limitations. First, it relies on pre-structured tables following the TableQA paradigm [Pasupat and Liang, 2015]; for tables embedded in unstructured text, existing text-to-table techniques [Wu et al., 2022, Deng et al., 2024] are required as a preprocessing step. Second, TAMO currently focuses on static table understanding in a single-turn setting. Extending it to support dynamic multi-step reasoning, complex table editing, and multi-turn dialogue over tables remains an open challenge. Third, we differentiate our work from layout-aware models such as DocLLM [Wang et al., 2024a, Liao et al., 2025]. These models are designed to understand tables presented as *images* within documents, focusing on layout and OCR challenges. TAMO addresses a different and complementary problem: reasoning over *pre-structured* tabular data. Fourth, our experiments utilized a consistent serialization format. A systematic study of how different text serialization templates (e.g., markdown vs. SQL-based) interact with our structural modality remains an important direction for future work. Finally, extensive multimodal instruction data is required to develop robust, out-of-the-box multimodal capabilities, which we leave for future work. These limitations highlight the early stage of our research and the need for further exploration to fully integrate table modalities with LLMs.

## 6  Conclusion

In this work, we introduced a novel framework, TAMO, which leverages a hypergraph-enhanced table encoder to boost frozen-parameter LLMs' generalization on tabular data. By adhering to the principle of table structure permutation invariance, TAMO effectively encodes table structures into LLM-comprehensible representations using learnable features. This enables the handling of tasks involving both text and table understanding, such as table QA. Additionally, we presented StructQA, a dataset focused on table structure understanding, and validated our framework's efficacy and versatility across four other public table QA benchmarks.

## Acknowledgments

This paper is supported by the National Regional Innovationand Development Joint Fund (No. U24A20254). Haobo Wang is also supported by the Fundamental Research Funds for the Central Universities (No. 226-2025-00004) and Zhejiang Provincial Universities (No. 226-2025-00065). This work was supported by Ant Group.

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

# A  StructQA Dataset Details

As mentioned in Section 3.1, we construct a table structure understanding dataset **StructQA**, which has 5 types of table structure tasks. Here, we provide the construct details. Specifically, we randomly select 500 tables from WikiTQ [Pasupat and Liang, 2015], creating 3 question templates for each table per task, resulting in 7500 question-answer pairs. We chose tables from WikiTQ as the source for StructQA due to its large, diverse collection of high-quality, real-world tables. Crucially, these tables are structurally simple for humans to understand. The fact that modern LLMs struggle with these seemingly simple structures makes their failure more pronounced and powerfully demonstrates the core problem we aim to solve. We split the data into training, validation, and test sets with a ratio of 60%, 20%, and 20%, respectively. The question templates for each task are as follows:

(1) *Cell location*

• What is the value in the column {column name} of sample row {row number}?

• Can you tell me the value of the column {column name} in sample row {row number}?

• In sample row {row number}, what is the value for the column {column name}?

(2) *Column lookup*

• In sample row {row number}, which columns contain the value {cell value}?

• Can you identify the columns in sample row {row number} that have the value {cell value}?

• Which columns in sample row {row number} are associated with the value {cell value}?

(3) *Row lookup*

• Which rows in the column {column name} have a value of {cell value}?

• Can you identify the sample rows where the column {column name} equals {cell value}?

• In the column {column name}, which rows contain the value {cell value}?

(4) *Column comprehension*

• What are the distinct values in the column {column name}?

• Could you list the unique values present in the column {column name}?

• In the column {column name}, what various values can be found?

(5) *Row comprehension*

• What are the values of each cell in row {row number} of the sample?

• Could you provide the cell values for each column in sample row {row number}?

• In sample row {row number}, what are the respective cell values?

# B  Public Datasets Details

**HiTab** [Cheng et al., 2022] is a table question answer dataset on *hierarchical* tables, which have a multi-level structure in the table header. It comprises 10,672 questions over 3,597 hierarchical tables. We use execution accuracy as the evaluation metric. Experiments on HiTab can effectively demonstrate the superiority of hypergraphs in modeling arbitrary hierarchical tables.

**WikiTableQuestions** (WikiTQ) [Pasupat and Liang, 2015] is a large-scale dataset for *complex* question answering over tables. It consists of 22,033 questions over 2,108 Wikipedia tables. The questions often require *complex reasoning and aggregation* operations. The primary evaluation metric for WikiTQ is the accuracy of the predicted answers compared to the ground truth.

**WikiSQL** [Zhong et al., 2017] is a dataset designed for natural language to SQL query generation. It contains 80,654 natural language questions paired with SQL queries and their corresponding answers over 24,241 tables from Wikipedia. We use the execution accuracy (the correctness of the query results) as the evaluation metric for WikiSQL.

*FeTaQA* [Nan et al., 2022] is a *free-form* table question answering dataset that emphasizes generating *comprehensive*, *free-text* answers. It comprises 10,279 questions over 3,641 tables, primarily sourced from Wikipedia. BLEU metric [Papineni et al., 2002] is recommended officially to evaluate the similarity between generated and reference answers.

# C  Experiments

## C.1  Implementation Settings

Experiments are conducted using 2 NVIDIA A100-80G GPUs. All experiments are conducted using the same set of random seeds for reproducibility.

**The table encoder.** We set the hidden dimension of the table encoder to 768 and use a 3-layer hypergraph transformer to model global table structure. To initialize each cell's semantic representation before hypergraph construction, we adopt the embedding of ''[CLS]'' token output from the pretrained RoBERTa-base model applied to the cell's textual content. The alignment projector is implemented as a single-layer MLP, which maps the encoder outputs to the LLM embedding space.

**LLMs.** We use the open-sourced Llama2-7b[9] as the LLM backbone. In fine-tuning the LLM with LoRA, the lora_r parameter (dimension for LoRA update matrices) is set to 8, and the lora_alpha (scaling factor) is set to 16. The dropout rate is set to 0.05. In prompt tuning, the LLM is configured with 8 virtual tokens. The number of max text length is 1024. The number of max new tokens, i.e., the maximum number of tokens to generate, is 128. We use Mistral-7B[10] for some experiments.

**Optimization.** We use the AdamW optimizer. We set the initial learning rate at 1e-5, with a weight decay of 0.05. The learning rate decays with a half-cycle cosine decay after the warm-up period. The batch size is 8, and the number of epochs is 10. To prevent overfitting and ensure training efficiency, an early stopping mechanism is implemented with a patience setting of 3 epochs.

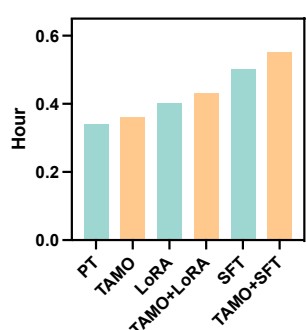

Figure 7: Training time efficiency comparison under different settings for 1 epoch on WikiTQ dataset.

## C.2  Efficiency Analysis of TAMO

To further demonstrate the practicality of TAMO, we evaluate its operational efficiency. In our experiments, we utilize a server equipped with 2 A100 GPUs. Only SFT uses 2 GPUs while conducting all other experimental setups with single GPU training. We measure the time required to run 1 epoch on the WikiTQ dataset. The results are shown in Figure 7. We found that (i)-TAMO has a faster runtime efficiency compared to LoRA; (ii)-TAMO$^{+}_{LoRA}$ shows only a slight increase in runtime compared to LoRA, as does TAMO$^{+}_{SFT}$ compared to SFT. Therefore, injecting learnable table features does not significantly add to the computational burden in practical applications.

We clarify that our "cost-effective" claim is relative to the computational cost of fine-tuning the whole LLM itself. As shown in Figure 7, TAMO (in the frozen LLM setting) requires significantly less training time than LoRA or full SFT, as we only train the lightweight table encoder and alignment projector.

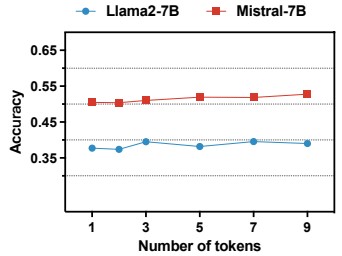

Figure 8: Analysis study of different numbers of table structure tokens on the WikiTQ dataset.

---

[9]https://huggingface.co/meta-llama/Llama-2-7b-hf
[10]https://huggingface.co/mistralai/Mistral-7B-v0.1

## C.3 Ablation of Table Token Number Parameter

We further explore the impact of the table structure token quantity parameter on the model's performance. Specifically, in the frozen LLM setting, we evaluate TAMO on the WikiTQ dataset with varying numbers of table structure tokens. Due to limited computational resources, we randomly select 6000 samples from the WikiTQ training set for the experiments, keeping the validation and test sets unchanged. The experimental results are shown in Figure 8. The final performance of the model is consistently similar when the number of tokens is two or more $\{2, 3, 5, 7, 9\}$, which indicates that a minimum of one token is sufficient to explain the structural information in the table.

## C.4 Ablation Study on Modality Components

To rigorously assess the contribution of each input modality in TAMO, as discussed in Section 2.3, we conducted an ablation study comparing three configurations: (1) **Graph-only**, using only the structural embeddings $\mathbf{X}_{st}$; (2) **Text-only**, using only the serialized text $\mathbf{X}_{tt}$; and (3) **TAMO**, our full multimodal approach combining both $\mathbf{X}_{st}$ and $\mathbf{X}_{tt}$. We evaluated these across all tuning settings (PT, LoRA, SFT) on the StructQA benchmark.

The results are presented in Table 4. The *Graph-only* pipeline consistently fails on this generative QA task (e.g., accuracy below 8% across settings), confirming that structural information alone is insufficient without semantic grounding. The *Text-only* model performs substantially better but exhibits lower accuracy and robustness compared to the full model, especially in parameter-efficient settings (PT, LoRA). Only the complete **TAMO** framework achieves the highest performance and robustness, empirically validating the synergistic necessity of combining both structural and textual modalities.

| Tuning | Modality | StructQA | Permute | Robustness |
|--------|----------|----------|---------|------------|
| PT | Graph-only | 7.00 | 7.13 | 14.60 |
| | Text-only | 37.80 | 29.93 | 31.07 |
| | TAMO (Both) | 59.07 | 43.47 | 43.80 |
| LoRA | Graph-only | 7.93 | 7.00 | 14.80 |
| | Text-only | 45.93 | 35.87 | 39.67 |
| | TAMO (Both) | 67.67 | 42.77 | 53.73 |
| SFT | Graph-only | 6.93 | 6.33 | 16.53 |
| | Text-only | 62.00 | 42.86 | 51.53 |
| | TAMO (Both) | 69.40 | 43.74 | 64.07 |

Table 4: Ablation on Modality Components (StructQA - Accuracy / Permutation Acc. / Robustness)

## C.5 Comparison with Structure-Aware Baselines (TAPAS/TAPEX)

To further situate TAMO relative to prior paradigms focused on table encoding, we conducted a direct comparison against established structure-aware encoder models, TAPAS [Herzig et al., 2020] and TAPEX [Liu et al., 2022], on our StructQA benchmark (Table 5). TAMO dramatically outperforms these baselines on the core structural understanding tasks (StructQA and Permutation Accuracy), showcasing the advancements achieved by integrating structural information directly into modern LLMs.

We observe that while TAPAS and TAPEX show higher robustness scores, this is largely an artifact of their very low base accuracy (e.g., 4.67% for TAPAS). It is easier to maintain consistency when a model is consistently incorrect. TAMO's performance demonstrates a far superior grasp of table structure.

## C.6 Evaluation of Cross-Dataset Generalization of TAMO

In Table 2, we demonstrated that TAMO, when trained individually on each dataset, achieves significant improvements on the corresponding test sets. This raised the question of whether TAMO's

| Model | StructQA (Acc.) | Permute (Acc.) | Robustness |
|---|---|---|---|
| TAPAS | 4.67 | 4.67 | 65.47 |
| TAPEX | 12.33 | 9.60 | 54.73 |
| TAMO (PT) | 59.07 | 43.47 | 43.80 |

Table 5: Comparison with TAPAS/TAPEX on StructQA

table structure embeddings are generalizable to other datasets. To address this, we evaluated TAMO models trained on one dataset against the test sets of other datasets, as shown in Table 6.

Theoretically, TAMO's table structure embeddings are designed to model general table structures. However, the training process also relies on task-specific instruction data, and the loss for learning table structure representations is tied to QA objectives. ***This means the table embeddings can be influenced by the types of instructions used during training, introducing task-specific biases.*** For example, embeddings trained on StructQA, which involves simpler table structures, tend to perform well on structural recognition tasks but lack the complexity required for reasoning-heavy tasks like WikiTQ. Consequently, while table structure embeddings trained on individual tasks consistently outperform baselines without structure embeddings, they fall short of matching the performance of embeddings trained directly on the target task. We also observed that datasets with significant differences, such as FetaQA—which uses BLEU as an evaluation metric for free-text answers—show limited cross-dataset transferability. The model trained on FetaQA fail to provide improvements on other datasets, and vice versa. However, for QA datasets with similar formats and objectives, such as WikiTQ and WikiSQL, we observed some degree of transferability, suggesting that TAMO can leverage shared patterns among related tasks. These observations are consistent with findings in TableLlama [Zhang et al., 2023b], where differences in task formats and reasoning complexity limited cross-task generalization.

| Evaluation Dataset | StructQA | HiTab | WikiTQ | WikiSQL | FetaQA |
|---|---|---|---|---|---|
| Metric | Accuracy | Accuracy | Accuracy | Accuracy | BLEU |
| Base | 8.60 | 7.77 | 14.50 | 21.44 | 20.08 |
| StructQA | **59.07** | 16.73 | 18.74 | 32.57 | 8.38 |
| HiTab | 17.53 | **48.86** | 27.46 | 38.83 | 1.78 |
| WikiTQ | 16.40 | 29.29 | **37.06** | 38.74 | 0.95 |
| WikiSQL | 18.73 | 24.43 | 23.85 | **76.45** | 1.18 |
| FetaQA | 0.00 | 0.00 | 0.02 | 0.00 | **36.52** |

Table 6: Generalization results of each TAMO separately trained on different dataset.

To isolate the effect of table structure representations from task-specific biases, we conducted additional experiments focusing solely on table structure prediction tasks. As shown in Table 3, table encoder trained on one dataset achieved F1 scores above **60%** on structure prediction tasks from the other dataset. This demonstrates that TAMO's table encoder captures a unified representation of table structures and validates the generalizability of our approach.

A key factor is the absence of large-scale, task-agnostic pretraining for TAMO's table encoder. Similar to how CLIP [Radford et al., 2021] decouples modality-specific representations through extensive pretraining, a dedicated pretraining phase for TAMO's table encoder—focusing purely on table-related structural information—could mitigate task-specific biases. This remains an important direction for future work to enhance generalization across domains and datasets.

### C.7 Effectiveness on Multiple-Table Scenarios

To validate TAMO in multiple-table scenarios, we have conducted additional experiments on the MultiTabQA-geoQuery [Pal et al., 2023] dataset. This dataset involves multiple-table queries with total token lengths reaching up to 4K, relatively larger than current TableQA benchmarks. Specifically, we evaluated its cell selection task using precision, recall, and F1 score as metrics. Due to the

| Setting | Method | Precision | Recall | F1 score |
|---|---|---|---|---|
| Inference Only | One-shot | 9.68 | 5.96 | 7.38 |
| Frozen LLM | Prompt tuning | 4.83 | 3.46 | 4.03 |
| | **TAMO** | 6.82 | 4.86 | 5.67 |
| | $\triangle_{Prompt\ tuning}$ | ↑ 41.20% | ↑ 40.46% | ↑ 40.69% |
| Tuned LLM (LoRA) | LoRA | 30.56 | 10.30 | 15.41 |
| | **TAMO**$^{+}_{LoRA}$ | 28.32 | 10.67 | 15.50 |
| | $\triangle_{LoRA}$ | ↑ −7.33% | ↑ 3.59% | ↑ 0.58% |
| Tuned LLM (SFT) | SFT | 30.55 | 11.04 | 16.22 |
| | **TAMO**$^{+}_{SFT}$ | **49.36** | **25.46** | **33.59** |
| | $\triangle_{SFT}$ | ↑ 61.57% | ↑ 130.62% | ↑ 107.09% |

Table 7: Effectiveness on MultiTabQA-geoQuery.

unique output format requirements of this task, we adopted a one-shot setting across the following experiments while keeping other parameters unchanged. As shown in Table 7, TAMO achieves over 40% and 100% improvements under frozen LLM and SFT LLM settings, respectively, demonstrating its effectiveness in multi-table scenarios. While TAMO shows only marginal advantages in the LoRA setting, we will investigate the detailed configurations in future work.

## C.8 Effectiveness for Different LLMs

In our previous experiments, we choose Llama2-7B as the baseline for a fair comparison with prior works. However, it is undeniable that, as a new modality, TAMO should theoretically enhance the performance of various LLMs. In this section, we examine the issue of the baseline. Briefly, we introduce three advanced baselines—TableLlama [Zhang et al., 2023b], Mistral-7B, and LLaMA 3.1 8B—and conducted experiments under the frozen LLM setting.

Table 8 and Table 9 demonstrate that: (i)-The minimal gap (0.0016 acc.) between the base and prompt tuning on TableLlama indicates that the supervised fine-tuned LLMs already possess a strong capability to follow tabular format instructions. Consequently, prompt tuning has a limited effect. However, *incorporating global table structure information through* **TAMO** *further enhances table reasoning capabilities.* (ii)-The ultimate performance of TAMO is influenced by the capability of the LLMs. For instance, Llama3 shows significantly better performance than TableLlama (based on Llama2). (iii)-while LLaMA 3.1 8B achieves a stronger baseline than LLaMA 2 7B, adding the table encoder consistently improved performance, with gains reaching over 10% on certain datasets. This further validates the unique benefits of hypergraph-based structural representation of tables across more advanced open-source LLMs.

| Method | Llama2 | TableLlama | Mistral |
|---|---|---|---|
| Inference Only (Base) | 14.50 | 31.22 | 18.44 |
| Prompt tuning | 29.86 | 31.38 | 44.98 |
| **TAMO** | **37.06** | **39.85** | **47.33** |
| $\triangle_{Prompt\ tuning}$ | ↑ 24.11% | ↑ 26.99% | ↑ 5.22% |

Table 8: Evaluate the scalability for different LLMs of our proposed TAMO on the frozen LLM setting (prompt tuning) on the WikiTQ dataset.

| Setting | Dataset
Task Type
Evaluation Metric | StructQA
Structural QA
Accuracy | HiTab
Hierarchical QA
Accuracy | WikiTQ
Table QA
Accuracy | WikiSQL
Table QA
Accuracy | FetaQA
Free-form QA
BLEU |
|---|---|---|---|---|---|---|
| Inference Only | Llama 3.1 8B | 15.73 | 19.51 | 23.80 | 31.60 | 14.05 |
| Frozen LLM | Prompt tuning
**TAMO**
$\triangle_{Prompt\ tuning}$ | 71.53
78.00
↑ 9.05% | 69.38
73.73
↑ 6.27% | 53.71
56.93
↑ 6.00% | 77.06
85.44
↑ 10.87% | 36.16
38.09
↑ 5.34% |

Table 9: Results on advanced LLM.

# D Discussions

## D.1 Positioning of TAMO

While both HyTrel [Chen et al., 2024] and TAMO adopt a hypergraph-based framework, there are significant distinctions. HyTrel focuses on general tabular representation learning and, as stated in its limitations, cannot handle joint text-table reasoning tasks like TableQA. In contrast, it is non-trivial for TAMO to pioneer treating tables as an independent modality within LLMs, aligning hypergraph-based table representations with text representations to tackle complex reasoning tasks.

This distinction parallels advancements in other domains. For example, in vision, ViT [Dosovitskiy et al., 2020] and CLIP [Radford et al., 2021] act as modality encoders, while GPT-4v [OpenAI, 2023] and LLaVA [Liu et al., 2023] integrate these encodings into multimodal frameworks. In the audio domain, there is a similar phenomenon, as shown in Table 10. For the first time, TAMO fills this gap in the table domain, going beyond a table encoder to a multimodal reasoning framework. This cross-modal fusion makes TAMO a significant advancement, not an incremental improvement. Notably, while TAMO and HyTrel share a similar network architecture, their training tasks and optimization objectives are entirely different, further underscoring the contribution of our approach.

| Domain | Modality Encoder | Multimodal LLMs |
|---|---|---|
| Vision Domain | ViT [2020], CLIP [2021] | GPT-4v [2023], LLaVA [2023], MiniGPT-4 [2023a] |
| Audio Domain | Whisper [2023] | SpeechGPT [2023a], AudioPaLM [2023] |
| Table Domain | HyTrel [2024] | **TAMO (Ours)** |
| Role | Encoding domain-specific data | Modality alignment with LLMs to obtain corresponding domain-specific multimodal models |
| Ability for Generative Tasks (e.g., QA) | No | **Yes** |

Table 10: Positioning of TAMO in the table domain.

## D.2 Comparison with Potential Approaches

We acknowledge that there are several alternative approaches to incorporating table structure into LLM-based models. These include: (1) using 2D positional embeddings to capture row and column information, (2) data augmentation techniques to enforce permutation invariance, and (3) injecting fine-grained structural representations tailored to specific downstream tasks. Below, we discuss the applicability and limitations of these strategies in contrast to our proposed approach.

**Using 2D positional embeddings to capture row and column information.** Using 2D positional embeddings is indeed a natural approach, as it captures row and column information directly. However, implementing this method often requires intrusive modifications to the position encoding layer of LLMs (e.g., as in TableFormer [Yang et al., 2022]), demanding extensive re-training of these position encodings. Such re-training is highly dependent on specific LLM architectures, and the learned modifications are not theoretically transferable to other LLMs. In contrast, our proposed table encoder is designed to **operate as an external plugin of tabular modality, minimizing modifications to the LLM itself.**

**Data augmentation techniques to enforce permutation invariance.** While data augmentation techniques to enforce permutation invariance are intuitive, they present practical challenges. For tables with dimensions $n \times m$, the number of possible permutations grows factorially as $n! \times m!$. Training on such a large augmented dataset is computationally prohibitive, and the resulting models are prone to overfitting due to the enormous training data requirements. TAMO is designed to **be data-efficient, achieving structural permutation invariance without relying on large-scale data augmentation.**

**Injecting task-specific fine-grained structural representations.** Several recent parallel studies have proposed injecting fine-grained table semantics into LLMs for specific structured reasoning tasks. For example, TNT [Long et al., 2025] targets the TEXT-TO-SQL task and models the database schema using column embeddings, enabling the LLM to generate accurate SQL queries. However, it ignores row-level semantics, making it less suitable for tasks like TABLEQA. HeGTa [Jin et al., 2025] adopts a heterogeneous graph encoder to capture rich structural features in complex tables (e.g., merged cells), but its graph encoding depends on the original table layout and lacks permutation invariance, making it sensitive to row/column reordering. Recent LLaSA [Xu et al., 2025] introduces a parameter-heavy G-Former module to unify various structured data formats (e.g., knowledge graphs, databases), yet it is primarily tailored for clean, schematic data and struggles with the irregular formats in real-world tables such as HiTab. Moreover, these methods typically require extensive pretraining. Concurrently, HyperG [Huang et al., 2025] mitigates sparsity via LLM-guided contextual augmentation, followed by semantic hypergraph construction and prompt-attentive hypergraph learning, which typically entails substantial extra preparation beyond task-level supervision due to the explicit augmentation pipeline. In contrast, our proposed TAMO framework is designed to be **pretraining-free and plug-and-play**, requiring only task-level training data for effective adaptation.

### D.3 Beyond Row and Column Permutations

While row and column permutations are the most prominent cases in tabular data, other forms of order permutations can arise in more complex table structures. These include:

**Nested Table Structures.** In hierarchical or grouped tables, sub-tables are often nested within a broader table structure. Permutations can occur within these nested sub-tables, reflecting changes in the ordering of hierarchical levels. Such structures are common in multi-level reports and datasets with grouped summaries.

**Composite Attributes.** Tables may contain multi-column attributes where relationships or dependencies exist between columns. For instance, in a table representing geographic data, attributes such as latitude and longitude might form a composite structure. Permutations within such attributes could represent alternative orderings of these dependent fields, requiring specialized handling to maintain semantic coherence.

**Cell-Level Permutations.** In some cases, individual cells may contain structured or semi-structured data, such as lists, arrays, or key-value pairs. Order changes within these cell values represent another form of permutation, particularly relevant in domains where embedded structured data is prevalent (e.g., JSON-like entries or lists of items within a cell).

While these forms of permutations are significant in certain contexts, they are most commonly observed in complex hierarchical datasets, such as HiTab [Cheng et al., 2022]. In this study, we focus primarily on flat table structures from mainstream TableQA datasets, where row and column permutations are the predominant concerns. Addressing these additional forms of permutation is an important direction for future work, particularly for datasets with more complex organizational patterns.

