# OpenReview forum: "Table as a Modality for Large Language Models"
_NeurIPS.cc/2025/Conference — NeurIPS 2025 poster_

### Official Review · Reviewer_vZ6u · 2025-06-15

**Clarity:** 2
**Significance:** 3
**Originality:** 2
**Rating:** 5
**Confidence:** 4

**Summary:**

This paper reports an approach of treating tables as a separate modality - so tables are encoded using a hyper graph transformer with multi set semantics to handle permutation invariance,  and then 'fused' with the text representation of the table using standard multi modal architectures to perform better table QA.

**Questions:**

1.  Use of the inference only baseline to make all comparisons seemed a bit weak to me - was this a few shot prompt case?  Were some structural examples provided?
2.  What exactly are the specialist SOTA methods - why do they not constitute a baseline and why is the fact that their performance better in those papers ignored?  Also, why aren't they evaluated on StructQA?  If the point is that these specialize SOTA methods (maybe refer to them by the citation in the table?) were specialized for the task - we do not know that do we we?  As in, those methods have not been evaluated on all the datasets mentioned here?


Thank you for addressing my questions - I believe you have addressed them satisfactorily.

**Ethical Concerns:**

["NO or VERY MINOR ethics concerns only"]

**Final Justification:**

I considered the response and I find answers to my questions satisfactory.

**Limitations:**

Yes

**Quality:**

2

**Strengths And Weaknesses:**

Strengths:
1.  The use of hyper graphs as table encoders is interesting, as is the use of pretraining autoregressive tasks that teach the model about the structure of tables.
2.  The results seem interesting although I do have some questions about the baselines.

Weaknesses:
1.  Use of the inference only baseline to make all comparisons seemed a bit weak to me - was this a few shot prompt case?  Were some structural examples provided?
2.  What exactly are the specialist SOTA methods - why do they not constitute a baseline and why is the fact that their performance better in those papers ignored?  Also, why aren't they evaluated on StructQA?  If the point is that these specialize SOTA methods (maybe refer to them by the citation in the table?) were specialized for the task - we do not know that do we we?  As in, those methods have not been evaluated on all the datasets mentioned here?
3.  If the point also is that this approach is more 'general' somehow - I'd question that as well because the table does need to be encoded by a specialized network.

---

> ### Author Rebuttal · Authors · 2025-07-31
>
> We thank the reviewer for their feedback and for finding our hypergraph-based approach interesting.
>
> **1. W1 / Q1 (Weakness of "Inference only" baseline):**
>
> The reviewer is correct: "Inference only" is a zero-shot baseline, chosen deliberately. First, there is no universally adopted few-shot methodology for the TableQA domain. Unlike other NLP tasks, each sample consists of a unique table and question. Constructing a few-shot prompt would require including multiple, often large, tables as examples, which is frequently impractical due to the context window limitations of most models.
>
> Second, the zero-shot baseline probes out-of-the-box LLMs’ intrinsic structural understanding, especially state-of-the-art table reasoners like TableLlama. Our results confirmed that even such models struggling here reveals a fundamental limitation, making this baseline a key diagnostic, not just a performance floor.
>
> Finally, our core improvement claims rely on comparisons with stronger trained baselines (Prompt Tuning, LoRA, SFT), where TAMO shows significant advantages (Table 2). We will label "Inference only" as "zero-shot" in revisions for clarity.
>
> **2. W2 / Q2 (Clarification of "Specialist SOTA" and evaluation on StructQA):**
> This is an excellent point. The term is indeed vague. In our revision, we will update the caption of Table 2 to explicitly name the specific model and provide its citation for each dataset. The specific models are:
>
> - For **HiTab**, the Specialist SOTA is **TableLlama**, which is based on Llama2-7B and underwent extensive supervised fine-tuning (SFT) on the HiTab training set along with other table QA datasets. The score reported in the original paper is 64.71, which we reproduced as 63.76.
>
> - For **WikiTQ** and **FetaQA**, the SOTA is **CABINET**, which improves LLM performance by using an external scorer to assign higher weights to question-relevant table cells, thereby filtering out noisy ones.
>
> - For **WikiSQL**, the SOTA is **SeaD**, which enhances text-to-SQL capabilities through a schema-aware denoising objective.
>
> It is important to note that our purpose in including these specialist SOTAs is to highlight different methodological directions. Unlike our TAMO framework, which directly enhances the LLM's end-to-end understanding of the table as a modality, methods like CABINET and SeaD rely on external modules or task-specific pre-processing.
>
> These models were not evaluated on our new StructQA benchmark because they often involve complex, dataset-specific pipelines that are not readily adaptable to a new task format without significant engineering effort. Our goal with StructQA was to test the core structural understanding of more generalist LLM-based approaches.
>
> **3. W3 (Generality of the Approach):**
>
> This is a fair point about the term "general." We use it in two senses:
>
> 1) **Generalization across structural variations**, such as the row and column permutations tested in StructQA, where TAMO shows superior robustness (Figure 6) .
>
> 2) **Architectural generality**, as TAMO is a "plug-and-play" module that can be integrated with various mainstream LLMs (e.g., Llama2, Mistral, Llama 3) without intrusive architectural modifications . This contrasts with methods that require re-training an LLM's internal position encodings.
>
> We will refine our language in the paper to be more precise about these two aspects of generality.

---

> > ### Comment · Reviewer_vZ6u · 2025-08-03
> >
> > Perhaps I should have been clearer - would a small amount of fine tuning have improved performance on any standard LLM?  Did you need this graph representation and treatment of table as a modality?

---

> ### Author Response · Authors · 2025-08-04
>
> Thank you for the clarification. This is an excellent question that gets to the heart of our paper's contribution, and we appreciate the opportunity to highlight the key results that directly address it.
>
> Let us first briefly recap the core premise of our work. Our investigation began with a foundational finding: **serializing tables into text, while convenient, causes LLMs to lose a precise understanding of the table's structure.** This was validated in our StructQA benchmark, where even advanced models struggled with simple structural tasks like permutation. To address this information loss, we proposed **modeling the table's structure explicitly using a graph representation** and treating it as a distinct modality.
>
> You asked if a small amount of fine-tuning would have been sufficient to improve performance, and whether our graph representation was truly necessary. **This is precisely the central question our main experimental setup in our paper's Table 2 was designed to answer.** In Table 2, for each of the **three standard fine-tuning paradigms (Prompt Tuning, LoRA, and full Supervised Fine-tuning)**, we directly compared two settings: a **text-only baseline** (representing standard fine-tuning) versus our **TAMO framework** (which adds the table modality by graph).
>
> The results, summarized below, unequivocally show that while standard fine-tuning does improve performance, **it is not sufficient**. Across all datasets and all tuning methods, adding our TAMO modality provides a substantial additional performance gain.
>
> | Tuning | StructQA | HiTab | WikiTQ | WikiSQL | FetaQA |
> |---|---|---|---|---|---|
> | PT on text | 37.80  | 26.26  | 29.86  | 61.24  | 29.94  |
> | **PT on TAMO** | 59.07  | 48.86  | 37.06  | 76.45  | 36.52  |
> | LoRA on text | 45.67  | 50.76  | 37.13  | 57.10  | 35.80  |
> | **LoRA on TAMO** | 70.80  | 59.22  | 43.53  | 84.43  | 37.43  |
> | SFT on text | 62.73  | 54.80  | 43.28  | 79.86  | 37.37  |
> | **SFT on TAMO** | 71.60  | 63.89  | 45.81  | 85.90  | 39.01 |
>
> This direct, controlled comparison proves that the performance gains are not just from fine-tuning, but specifically from the structural information captured by our graph representation. Therefore, to answer your question directly: **yes, the graph representation and treatment of the table as a modality are indeed necessary to overcome the fundamental limitations of text serialization, a fact that standard fine-tuning alone cannot resolve.**
>
> Furthermore, to demonstrate the broad applicability of our findings, we have validated this conclusion across multiple LLM families. Our main paper shows these results on Llama2-7B and LLaMA 3.1 8B. We have since conducted additional experiments on the newer Qwen2.5-7B-Instruct and Qwen3-8B models, with the same consistent outcome:
>
> | Model | Setting | StructQA | Permutation | Robustness |
> |---|---|---|---|---|
> | Qwen2.5-7B | Inference | 38.20 | 31.80 | 40.87 |
> | Qwen2.5-7B | PT on text | 45.53 | 33.80 | 33.33 |
> | **Qwen2.5-7B** | **TAMO** | **72.27** | **61.33** | **62.73** |
> | Qwen3-8B | Inference | 46.86 | 41.73 | 48.87 |
> | Qwen3-8B | PT on text | 54.80 | 45.73 | 45.47 |
> | **Qwen3-8B** | **TAMO** | **70.20** | **53.93** | **55.60** |
>
> In summary, the evidence from multiple fine-tuning methods and across diverse LLM families robustly confirms that our TAMO framework is a necessary and highly effective solution.
>
> *We hope this detailed explanation, grounded in the core motivation and experimental design of our paper, fully addresses your concerns. Thank you again for your time and engagement.*

---

> > ### Comment · Reviewer_vZ6u · 2025-08-06
> >
> > Thank you for that explanation - I will adjust my score appropriately.

---

### Official Review · Reviewer_Qmvd · 2025-06-21

**Clarity:** 2
**Significance:** 3
**Originality:** 3
**Rating:** 4
**Confidence:** 4

**Summary:**

The paper introduces TAMO which emphasizes on treating tables as an independent modality for tabular Question-Answering (QA) tasks. The paper presents a novel instruction tuning paradigm by combining structural information in tables and semantic information in text towards augmenting the capabilities of a pretrained LLM towards robust QA. TAMO further introduces the StructQA benchmark which evaluates various impending challenges like permutation invariance, structural variations etc. important for downstream tabular learning tasks.

**Questions:**

1. What is the importance of using only WikiTQ over other tabular datasets in StructQA ?

2. Does the experimental results of TAMO depend on the serialization template used ? It would be good to know the variation in performance when using direct serialization as in TAPAS, SQL based serialization in TAPEX etc.

3. It is not clear whether a graph-only pipeline would be sufficient for the improvements discussed in TAMO. Following Figure 6 where it is clear that text-only pipelines are insufficient the authors must include experiments comparing graph-only pipelines like GCondNet (Margeloiu et al., 2022), IGNNet (Alkhatib et al., 2024) etc. against TAMO. This is critical to establish the necessity of a multimodal model in TabularQA tasks.

4. TAMO utilizes the serialized tables as additional input in their pipeline. Does this indicate that during the hypergraph transformation the learnt representation loses semantic understanding of the table ?

**Minor Suggestions -**
1. The section on Hyper-Trans (Chen et al., 2021) (lines 133-152) is not unique to TAMO and should definitely be moved to the appendix. The authors should use this space to highlight the core differences between TAMO and other existing methods in Tabular QA.

2. The use of non-standardized mathematical notations in equations 1 through 3 makes the paper very hard to follow. I would request the authors to use meaningful notations and elucidate them in the text wherever possible. Eg. usage of $rFF$ in line 148 seems to be $r$ times $FF$, $X$ and $H$ in equation 3 was never defined etc.

**Ethical Concerns:**

["NO or VERY MINOR ethics concerns only"]

**Final Justification:**

From the paper contents and the discussion in the rebuttals I feel that the methodological contribution of TAMO is incremental and is better positioned as a multimodal instruction tuning strategy for Tabular QA. Nevertheless the authors have successfully demonstrated good empirical results in multimodal Tabular QA which is encouraging.

**Limitations:**

Yes, Section 5 of the paper does mention several limitations of the paper which are valid and encourages future research in this space.

**Quality:**

2

**Strengths And Weaknesses:**

**Strengths -**

1. TAMO introduces a novel instruction tuning architecture which combines text and hypergraph representations of tables towards Tabular QA tasks.

2. The introduction of StructQA benchmark to evaluate permutation and structural invariance - a necessary evil in tabular QA tasks is a key contribution.

**Weaknesses -**

1. To the best of my knowledge, the introduction of Multi-Modal graph and text based tabular learning is not novel to TAMO. An example would be TabGLM (Majee et al., 2025) which combines graph and textual representations from tables for auxiliary tasks. However, the introduction of a multimodal instruction tuning model for Tabular QA would be a more appropriate positioning for TAMO.

2. The claim in line 182 that TAMO is cost-effective may not be true or not clearly supported. The model in Figure 4 uses two encoders and an LLM instead of single encoders in unimodal pipelines like TAPAS (Herzig et al., 2020), TAPEX (Liu et al., 2022) ALTER (Zhang et al., 2024) and TaBERT (Yin et al., 2020) which might be more cost-effective than TAMO.

3. The experiments seem to be insufficient - TAMO (a specialized multimodal model) is contrasted against general purpose LLMs like GPT-3.5, 4 etc. without contrasting against several existing tabular LLMs like TabLLM, TAPAS, TAPEX, TaBERT etc. which have already been established for Tabular QA and argue that representing a table as a natural language can largely mitigate permutation invariance (a major pitfall analyzed in this paper).

---

> ### Author Rebuttal · Authors · 2025-07-31
>
> We thank the reviewer for their detailed feedback and for recognizing the value of our StructQA benchmark.
>
> **1. W1, W3 (Novelty and Insufficient Baselines):**
>
> We thank the reviewer for the reference to TabGLM. While prior works have combined graphs and text, TAMO's novelty lies in pioneering the treatment of table structure as a distinct modality within the modern large language model instruction-tuning paradigm. Our framework is designed as a general, plug-and-play module for various LLMs, which is a significant advancement. We will revise our introduction and related work to discuss TabGLM and further sharpen our positioning.
>
> Regarding baselines like TAPAS/TAPEX, we have run these experiments on our StructQA benchmark. The results are as follows:
>
> | Model | StructQA (Accuracy) | Permute (Accuracy) | Robustness |
> |---|---|---|---|
> | **TAPAS** | 4.67 | 4.67 | 65.47 |
> | **TAPEX** | 12.33 | 9.6 | 54.73 |
> | TAMO | 59.07 | 43.47 | 43.8 |
>
> The results clearly show that TAMO dramatically outperforms these baselines on the core structural understanding tasks (StructQA and Permutation Accuracy). It is crucial to consider the architectural differences: TAPAS is a BERT-based encoder, and TAPEX is a BART-based encoder-decoder model. In contrast, our TAMO framework is specifically designed as a multimodal extension for **decoder-only LLMs**, which represent the current dominant architecture. While TAPAS and TAPEX show higher robustness, this is largely an artifact of their very low base accuracy; it is easier to remain consistent when a model is consistently incorrect. TAMO's performance demonstrates a far superior grasp of table structure. We will add a new table and this analysis to the appendix in our revised manuscript.
>
> **2. W2 (Claim of "Cost-Effective"):**
>
> We appreciate the request for clarification. The "cost-effective" claim is relative to the cost of **fine-tuning the LLM itself**. As shown in our efficiency analysis in **Appendix C.2 (Figure 7)**, TAMO's training time (in the frozen LLM setting) is significantly less than LoRA or full supervised fine-tuning (SFT) because we only train the small encoder and projector.
> Furthermore, to clarify the architecture in Figure 4, while there are two input streams, our framework only introduces **one new encoder (the table encoder)**. The textual input is processed directly by the native embedding layer of the **decoder-only LLM**, which does not require a separate text encoder. This is fundamentally different from models like TAPEX or TaBERT, which employ dual-encoder architectures and are thus computationally more demanding. We will clarify this context in the main text.
>
> **3. Q1 (Why only WikiTQ for StructQA?):**
>
> We chose tables from WikiTQ because it offers a large and diverse collection of high-quality, real-world tables from Wikipedia. These tables are structurally simple for humans to understand. The fact that modern LLMs still struggle with these seemingly simple structures makes their failure more pronounced and powerfully demonstrates the core problem we aim to solve. This diversity and simplicity were essential for creating a benchmark that robustly tests a model's fundamental structural understanding. We will add this justification to Appendix A.
>
> **4. Q2 (Dependence on Serialization Template):**
>
> This is an insightful point. While we used a consistent format for fair comparison, our core hypothesis is that any serialization format inevitably leads to a loss of structural information. TAMO is designed to compensate for this fundamental loss, regardless of the specific template. A systematic study of different templates is an excellent direction for future work, and we will add this as a discussion point in our limitations section.
>
> **5. Q3 (Sufficiency of a Graph-only Pipeline):**
>
> We would like to clarify that a graph-only pipeline is not applicable for the tasks we address. Table question-answering requires understanding a **natural language question**. A pure graph-based model cannot process this textual query. The LLM is therefore an indispensable component for its language comprehension and reasoning abilities.
> Furthermore, we wish to clarify that the models mentioned by the reviewer, GCondNet and IGNNet, are designed for **tabular prediction tasks** (i.e., predicting a label for each row based on its features). This is a different task from **Table QA**, which involves generating a natural language answer based on a table and a question. Therefore, these models are not applicable baselines for our work.
>
> **6. Q4 (Hypergraph Losing Semantic Understanding):**
>
> This observation is correct and highlights the synergy of our approach. The hypergraph encoder is primarily designed to capture **structural and relational properties**. The serialized text, processed directly by the LLM, remains crucial for a deep, contextual understanding of the cell contents and the question. The two modalities are thus complementary.

---

> > ### Comment · Reviewer_Qmvd · 2025-08-04
> >
> > I would like to thank the authors for the detailed rebuttal. Some of my questions regarding additional baselines, cost-effectiveness etc. have been adequately answered. A few follow-ups -
> >
> > 1. I am still unclear on the motivation behind treating hypergraphs as a completely separate modality. Based on the description in Sec. 2.2 the method transforms the input table into a hypergraph (not novel to TAMO and defined in Hytrel (Chen et al., 2023)). Next, the table is encoded using equations 1,2,3 which also is not novel to TAMO. To the best of my knowledge, this complete process defines a way to encode tabular information into embeddings for downstream QA tasks and does not constitute a new modality. Can the authors clearly describe the fundamental difference between the encoding of tables in TAMO vs. Hytrel ?
> >
> > 2. In continuation to Q4 since encoding the table as a hypergraph loses semantic knowledge of the table contents, can the authors show the contrast between a graph-only (only hyper-graph based QA), text-only (use only serialized text for QA) and multimodal (the complete TAMO for QA) architectures ?  Fig.6 currently compares only text-only and TAMO but a graph-only pipeline would be an important addition.

---

> > > ### Author Response · Authors · 2025-08-05
> > >
> > > Thank you for the thoughtful follow-up. Your questions allow us to better articulate our core contribution, which we now clarify in two parts.
> > >
> > > **1. Our Contribution: A Modality Interface for Tables, Not Just Another Encoder**
> > >
> > > Our work introduces a **modality interface** for tabular data, inspired by multimodal LLMs like LLaVA [1] and Qwen-VL [2]. Just as images (non-textual information) are converted into embedding token sequences via vision encoders to condition the LLM's generation, we convert tabular structure into a sequence of embeddings that are aligned with the LLM's input space. These structure tokens are injected alongside serialized text and the question, thereby conditioning the full autoregressive distribution: $ \prod_i p(a_i \mid X_{\text{struct}}, X_{\text{text}}, Q, a_{<i})$.
> > >
> > > While our table encoder reuses the Hyper-Trans architecture from HyTrel, the difference lies in its purpose and usage: prior hypergraph encoders typically produce pooled embeddings for classification. In contrast, we emit a sequence of structural tokens aligned to the LLM and optimized for generative QA. As discussed in **Appendix D.1**, this mirrors how CLIP [3] / ViT [4] was repurposed in LLaVA [1]: *the encoder is reused, but the LLM-facing interface and objective unlocks new capabilities.*
> > >
> > > ---
> > >
> > > **2. Why Tables Deserve Structural Representation**
> > >
> > > We acknowledge the underlying question: why introduce a complex new encoding for tables, when they can simply be serialized into text? This seemingly counter-intuitive step is, in fact, the core of our proposal. Our motivation stems from a key finding from our StructQA benchmark: **we found that the current reliance on text serialization is insufficient**, as it consistently leads to the loss of critical structural information. We therefore respectfully wish to convey a new perspective: **tables can be encoded into a richer, structural embedding to complement the serialized text and mitigate this information loss**. This hypothesis directly drives our experimental design.
> > >
> > > To evaluate the *standalone* value of our structure encoder, **Section 3.6** demonstrates that it learns generalizable patterns: when trained on one dataset, it achieves over 60% F1 on structure classification tasks from another (vs. 5.39% for a plain MLP).
> > >
> > > However, *structure-only (or graph-only) QA* is intentionally ill-posed: the model lacks lexical grounding from the question and cell values. Our updated experiments reflect this:
> > >
> > > | Tuning | StructQA | Permute | Robustness |
> > > |---|---|---|---|
> > > | PT on graph | 7.00  | 7.13  | 14.60  |
> > > | PT on text | 37.8 | 29.93 | 31.07 |
> > > | PT on TAMO | 59.07 | 43.47 | 43.8 |
> > > | LoRA on graph | 7.93  | 7.00  | 14.80  |
> > > | LoRA on text | 45.93  | 35.87  | 39.67  |
> > > | LoRA on TAMO | 67.67  | 42.77  | 53.73  |
> > > | SFT on graph | 6.93  | 6.33  | 16.53  |
> > > | SFT on text | 62.00  | 42.86  | 51.53  |
> > > | SFT on TAMO | 69.40  | 43.74  | 64.07 |
> > >
> > > These results show:
> > > - Graph-only fails to align with LLMs due to lack of text, not due to poor structure encoding. This is because our current encoder was not designed to capture the full semantics of the table, as this was beyond the scope of our initial validation.
> > > - Text-only fails to capture structural invariance.
> > > - Only TAMO, our multimodal interface, consistently achieves strong and robust performance.
> > >
> > > This validates our design: the structure encoder is valuable, but must be properly aligned and injected via a modality interface to be effective in generative tasks. TAMO provides such an interface, and does so in **a plug-and-play fashion without altering the base LLM**.
> > >
> > > We see our work as a humble but important initial exploration to challenge an established convention. *We hope it inspires the community to consider new directions, such as how to build a future table encoder that can create a complete table embedding.*
> > >
> > > We sincerely thank you for your insightful questions, which have given us a crucial opportunity to better articulate our vision. We hope this clarification has fully addressed your concerns to your satisfaction.
> > >
> > > ---
> > >
> > > [1] Liu, Haotian, et al. "Visual instruction tuning." Advances in neural information processing systems 36 (2023): 34892-34916.
> > >
> > > [2] Bai, Shuai, et al. "Qwen2. 5-vl technical report." arXiv preprint arXiv:2502.13923 (2025).
> > >
> > > [3] Radford, Alec, et al. "Learning transferable visual models from natural language supervision." International conference on machine learning. PMLR, 2021.
> > >
> > > [4] Dosovitskiy, Alexey, et al. "An Image is Worth 16x16 Words: Transformers for Image Recognition at Scale." International Conference on Learning Representations. 2020.

---

> > > > ### Comment · Reviewer_Qmvd · 2025-08-07
> > > >
> > > > I would like to thank the authors for the detailed response to the additional questions. At first, my question (2) on the ablation on graph-only vs. text-only vs. multimodal (TAMO) has been fully addressed. I will strongly recommend the authors to add this discussion to the main paper in the next revision for improved understanding.
> > > >
> > > > I also completely understand the mechanism discussed in the paper and the rebuttal on interfacing tabular data with multi-modal LLMs. As mentioned by the authors in the earlier response the process of encoding the table in TAMO is very similar to the one in Hytrel (with an addition of a MLP layer eq. 4 acting as projection layers) which limits the novelty of the work.
> > > >
> > > > In its current form I feel that the paper is better positioned as a  multimodal instruction tuning model for Tabular QA. Nevertheless the instruction tuning results are encouraging and very positive. I will adjust my evaluation accordingly.

---

> > > > > ### Author Response · Authors · 2025-08-07
> > > > > **Thank you**
> > > > >
> > > > > Thank you so much for the kind words—your positive feedback truly means a lot to us and has been a great source of encouragement. We're very glad to hear that our clarifications helped address your concerns.
> > > > >
> > > > > We also want to sincerely thank all the reviewers for their thoughtful suggestions throughout the discussion. We’re committed to incorporating the insights from these rich discussions into the final version of our paper—including the new experiments and reorganized explanations—to enhance its clarity and ensure the community can better understand our work's context, strengths, and limitations.
> > > > >
> > > > > Thank you again.

---

### Official Review · Reviewer_7uvU · 2025-06-28

**Clarity:** 3
**Significance:** 3
**Originality:** 3
**Rating:** 5
**Confidence:** 4

**Summary:**

This paper addresses the topic of LLM-based question answering regarding tables.
It presents a new framework for this, TaMo, and shows that it outperforms previous methods, including using GPT4, GPT 3.5 and Llama 2 7B, and it is robust to permutations. The framework is based on a new method for representing complex tables as hypergraphs and encoding them. It then uses a fine-tuned Llama 2-7B model to answer questions based on the embeddings of both the structure and content of the table. The paper presents comparisons based on 5 question answering datasets, one of them, StructQA, was developed as a part of this work and includes 7500 examples based on 500 tables. The reported results are 42.65% average improvement in these comparisons (ranging from 4.5% to 86%), as well as outperforming the OpenAI models in 4 out of the 5 datasets. Further analysis in done on the changes in attention and the good generalization of the proposed method.

**Questions:**

Please perform a comparison with recent models, such as GPT4.1, O4-mini, Llama 3.3 and DeepSeek-R1.

**Ethical Concerns:**

["NO or VERY MINOR ethics concerns only"]

**Final Justification:**

I feel that the authors have addressed my requests and the concerns of other reviewers. They provided comparisons with state-of-the-art models that they will include in their final version. Although the analysis of tables may seem like a narrow problem, I think it is important from a practical perspective. I raised my score since I think that the technical contribution and its practical value are significant enough for publication in NeurIPS.

**Limitations:**

Yes

**Quality:**

3

**Strengths And Weaknesses:**

The paper addresses an important problem and suggests a new approach for handling it. It includes significant experimentation, and it also provides a new dataset. It is clearly written and original.
The main quality weakness in my view is that it compares with old models (from over a year ago), which also raises concerns about its current significance. To be considered for NeurIPS, it would be important to show comparisons using the current state of the art models, including GPT4.1, O4-mini, Llama 3.3 and DeepSeek-R1, to clarify whether this method is still useful after the many improvements in LLMs that took place in the recent year.

---

> ### Author Rebuttal · Authors · 2025-07-31
>
> We sincerely thank the reviewer for their valuable feedback and for acknowledging the importance of our work.
>
> **Comparison with Advanced Models:**
>
> We completely agree with the reviewer that evaluating on the latest models is crucial in this fast-paced field. To thoroughly address this critical point, we allocated significant additional computational resources to conduct comprehensive new experiments, including on several models accessible only via paid APIs.
>
> First, to test whether the problem we identified still exists in the latest models, we evaluated several state-of-the-art LLMs you suggested on our StructQA benchmark. The results are as follows:
>
> | Model | StructQA | Permutation | Robustness |
> |---|---|---|---|
> | **GPT-4.1** | 60.33 | 49.86 | 60.07 |
> | **O4-Mini** | 55.93 | 53.27 | 59.40 |
> | **Llama-3.3-70B-Instruct** | 53.93 | 48.60 | 52.26 |
> | **DeepSeek-R1** | 57.46 | 52.07 | 63.2 |
> | GPT-4 (Original) | 51.40 | 47.13 | 50.53 |
>
>
> These results are very telling. Even after more than a year of rapid progress, today's most powerful LLMs still struggle with table structure understanding, a task that is trivial for humans. While their performance on the original StructQA test set has improved (reaching ~60%), every single model shows a significant performance degradation after permutation. ***This confirms our central hypothesis: the serialization of tables into text sequences leads to a loss of critical structural information, a problem that persists even in the latest SOTA models.*** This finding significantly strengthens the motivation for our work.
>
> Second, to demonstrate that our proposed method, TAMO, remains effective on newer models, we conducted further experiments. Our original choice of a 7B-scale model was to ensure a fair comparison with the established TableLlama baseline. In addition to the LLaMA 3.1 8B results already in our **Appendix C.6**, we have now evaluated TAMO on the recent Qwen2.5-7B-Instruct and Qwen3-8B models. The results are compelling:
>
> | Model | Setting | StructQA | Permutation | Robustness |
> |---|---|---|---|---|
> | Qwen2.5-7B | Inference Only | 38.20 | 31.80 | 40.87 |
> | Qwen2.5-7B | Prompt Tuning | 45.53 | 33.80 | 33.33 |
> | **Qwen2.5-7B** | **TAMO** | **72.27** | **61.33** | **62.73** |
> | Qwen3-8B | Inference Only | 46.86 | 41.73 | 48.87 |
> | Qwen3-8B | Prompt Tuning | 54.80 | 45.73 | 45.47 |
> | **Qwen3-8B** | **TAMO** | **70.20** | **53.93** | **55.6** |
>
>
> As shown, TAMO provides a dramatic performance boost on these new models. For instance, on Qwen2.5-7B, TAMO improves the accuracy on StructQA from 45.53% (Prompt Tuning) to **72.27%**, an absolute improvement of nearly **27 points**. This clearly demonstrates that ***our method of treating tables as a distinct modality is a powerful and relevant solution for enhancing the structural reasoning capabilities of even the most recent LLMs.***
>
> We will prominently feature these new findings in the revised manuscript. Thank you again for pushing us to strengthen our work in this crucial dimension.

---

> ### Comment · Reviewer_7uvU · 2025-08-01
> **Follow up questions**
>
> Thank you for your detailed response and additional experimental results, comparing with current models. The additional comparisons are only provided for StructQA, the dataset that you developed, but not for other public datasets included in your paper. Would you be able to provide a comparison for at least a couple of the public datasets, comparing your method at least to GPT4.1? Presenting an improvement over current models only for one dataset that you developed seems less significant than showing it for some existing datasets.

---

> ### Author Response · Authors · 2025-08-03
>
> Thank you for this crucial follow-up; it's an excellent point that pushes us to clarify the significance of our work. We agree completely and have conducted the extensive benchmarking experiments you requested. We respectfully clarify that our original evaluation philosophy was not to claim our 7B model surpasses foundation models orders of magnitude larger, but rather to demonstrate two key points: (1) *structural reasoning challenges persist even in advanced models*, and (2) *our architectural innovation enables much smaller models to become highly competitive*.
>
> | Models | StructQA | HiTab | WikiTQ | WikiSQL | FetaQA |
> |---|---|---|---|---|---|
> | GPT 4.1 | 60.33  | 60.54  | 68.14  | 71.21  | 36.75  |
> | Llama-3.3-70B-Instruct | 53.93  | 59.72  | 59.86  | 66.92  | 42.21  |
> | DeepSeek-R1 | 57.47  | 63.89  | 75.76  | 71.91  | 13.10  |
> | **TAMO (SFT + Llama2-7B)** | **71.60**  | **63.89**  | 45.81  | **85.90**  | 39.01 |
>
> The results validate our approach. TAMO significantly outperforms all advanced models on StructQA (+11.27) and WikiSQL (+14.69), achieves state-of-the-art on HiTab, and remains competitive on FetaQA. WikiTQ's general-knowledge focus favors larger models' parametric knowledge, but TAMO's strong performance across four diverse benchmarks demonstrates broad applicability.
>
> We hope these comprehensive results address your concerns and demonstrate that our architectural innovation enables accessible models to achieve remarkable structural reasoning capabilities, highlighting the real-world value of our contribution. Thank you again.

---

> > ### Comment · Reviewer_7uvU · 2025-08-05
> > **Thank you**
> >
> > Thank you for your response, which answers my question well. I will wait for your response to other reviewers to better assess the significance and re-consider my score accordingly. Thank you.

---

> > > ### Author Response · Authors · 2025-08-05
> > >
> > > Thank you very much for your follow-up and for your patience. We have now posted our detailed public response to Reviewer Qmvd, which addresses the fundamental questions regarding our work's core contribution and methodology.
> > >
> > > We hope this comprehensive clarification is helpful for your final assessment. It is our sincere hope that our work, and the rich discussion it has generated with all reviewers, offers some valuable insights for both you and the broader TableQA and LLM community.
> > >
> > > Please let us know if anything remains unclear; we would be happy to discuss further. Thank you again for your engagement.

---

> > > > ### Comment · Reviewer_7uvU · 2025-08-05
> > > > **A clarification**
> > > >
> > > > Thank you, I read your response and I will consider it. I have a clarification question: in case the decision will change and the paper will be accepted, will you incorporate the results you provided me (checking new models on several datasets) into the final version of the paper itself?

---

> > > > > ### Author Response · Authors · 2025-08-06
> > > > >
> > > > > Thank you for this follow-up. Yes, absolutely.
> > > > >
> > > > > We are fully committed to integrating the additional results and clarifications provided during the review process into the final version of the paper. In particular, the new experiments involving recent foundation models across multiple datasets will be included in the main text and figures.
> > > > >
> > > > > More broadly, we view the review process as a constructive dialogue with the community. We will revise the paper to reflect the valuable insights from all reviewers, including restructuring for clarity, strengthening the positioning of our contributions, and expanding the discussion of limitations.
> > > > >
> > > > > We are deeply grateful for your engagement—your feedback has meaningfully shaped how we present and sharpen the ideas in this work. Please feel free to let us know if any aspect remains unclear—we’d be glad to elaborate.

---

### Official Review · Reviewer_v8d6 · 2025-07-02

**Clarity:** 3
**Significance:** 2
**Originality:** 3
**Rating:** 4
**Confidence:** 4

**Summary:**

This paper proposes a multimodal large language model called TAMO: by converting table contents into hypergraphs, then further transforming them into embeddings, which are fed into the LLM to enhance its overall understanding capabilities. Results show that this method can achieve significant improvements on multiple benchmarks. Besides, a novel tabular-structure-understanding benchmark is released for the community.

**Questions:**

Please refer to the Strengths and Weaknesses.

**Ethical Concerns:**

["NO or VERY MINOR ethics concerns only"]

**Final Justification:**

Thanks for solving my concerns and I hope this work is thoughtful for the community. I will raise my rating to 4

**Limitations:**

Please refer to the Strengths and Weaknesses.

**Paper Formatting Concerns:**

No formatting concerns.

**Quality:**

3

**Strengths And Weaknesses:**

Strengths:
1. The approach is innovative: For table understanding scenarios, this paper proposes a special encoding scheme for table reasoning tasks, which introduces a graph-based approach into the LLM to enhance model performance.

2. The logic is clear: The paper first highlights the importance of table representation, as shown in Fig. 2, then designs a specific table encoder to address issues in table understanding scenarios (Sec. 2), and demonstrates the benefits of this approach through experimental validation (Tab. 2).


Weaknesses:
1. As described in Sec. 2 & Figure 4, TAMO adds an additional multimodal processing component to represent the content of tables, while the original table content serialized as text is still preserved as input to the LLM. Based on experience, this approach essentially provides the LLM with detailed inputs from multiple perspectives, which naturally enables the model to achieve better performance. However, does this approach (keep the serialized table text) introduce redundancy in information representation? In other words, why does the parsed table text still need to be input into the LLM in text modality? Additionally, it is recommended to further clarify the statistics in Table 2 (Main results).

2. The experimental conclusions are not sufficient enough: Since the addition of new modalities on LLM is not a completely innovative solution, but rather a situation where ongoing work is continuously innovating solutions on this topic. As a new modality bridging solution, has it been better compared with other multimodal solutions, such as DocLLM (Layout-Aware) and QWEN2-VL / TextMonkey (Image-Aware)? Can further analysis of this part be provided in Section 3.3?

---

> ### Author Rebuttal · Authors · 2025-07-31
>
> We thank the reviewer for their positive feedback on our work's innovation and clear logic. We address the thoughtful questions below.
>
> **1. W1 (Information Redundancy of Using Both Serialized Text and Structure Embeddings):**
>
> This is an excellent question regarding our design philosophy. We retain the serialized text because it provides the **fine-grained semantic content** of the table, which LLMs are exceptionally skilled at processing. The hypergraph-based structure embedding, conversely, provides the **global relational structure** (e.g., row-column relationships, hierarchies) that is inherently lost during serialization.
>
> The two modalities are complementary: the text provides the content ("what"), and the structure embedding provides the context and relations ("where"). As our case study in Figure 5 suggests, the structure token helps the LLM better focus its attention on the relevant parts of the serialized text to locate the correct answer. Therefore, this dual representation is not redundant but synergistic.
>
> **2. W2 (Comparison with Other Multimodal Solutions like DocLLM):**
>
> We thank the reviewer for raising this point. We wish to clarify a fundamental distinction in the problem setting. Models like DocLLM or vision-language models (e.g., Qwen-VL) are designed to understand tables presented as images within documents, **focusing on layout and OCR challenges**. Our work, TAMO, addresses a different and complementary problem: reasoning over pre-structured tabular data (e.g., from databases, CSV files). As stated in our Limitations (Section 5), TAMO operates on this structured data format. The modalities are different (image+text vs. graph_structure+text), and the core tasks are distinct. Therefore, a direct comparison would not be appropriate.

---

> > ### Author Response · Authors · 2025-08-06
> > **Follow-up by Authors**
> >
> > Dear Reviewer,
> >
> > We are writing to follow up on our rebuttal and hope that we have comprehensively addressed your initial questions and concerns.
> >
> > Several rounds of public discussion with other reviewers have helped clarify our contributions and methodology further; we believe they may also address potential follow-up concerns from your side.
> >
> > In short, our work introduces a new perspective to the TableQA and LLM communities. Beyond the standard practice of representing tables via text serialization, we propose using table embeddings to provide the LLM with more comprehensive and robust structural understanding. We hope this initial exploration can inspire the community to think beyond the limitations of text-only table representations.
> >
> > We truly value this discussion process and are committed to incorporating insights from all reviewer conversations into the final version. We believe this will greatly help readers better understand our work's context, strengths, and limitations.
> >
> > We hope our clarifications have been satisfactory. If you have any further questions, please do not hesitate to let us know—we would be happy to discuss them.

---

> > ### Comment · Reviewer_v8d6 · 2025-08-08
> >
> > Thank the authors for the detailed reply. Regarding Q1, my confusion is whether the textual content of the table is already present within the Unified Hypergraph Modeling; if so, does this become redundant with the step “Serialize Table into Formatted Text Sequence”? The description of this design in the paper is not very clear, and I believe it is necessary to further elaborate on it.
> >
> > Regarding Q2, my main concern is whether this way of constructing the modality offers clearer comparative advantages or disadvantages relative to other table representation approaches; clarifying this would help everyone understand it more fully.

---

> > > ### Author Response · Authors · 2025-08-08
> > >
> > > Thank you for the detailed follow-up. These are excellent questions that allow us to clarify the core mechanics and advantages of our approach.
> > >
> > > **1. Q1 – On redundancy:**
> > >
> > > Our hypergraph encoder does not carry the full semantic content of the cells. While we use cell text to initialize node embeddings, the encoder’s purpose is to abstract **structural relations** (row/column alignment, hierarchies) rather than learn rich semantics. The serialized text is therefore still essential to provide the **lexical grounding** for answering the question. The two inputs are complementary: text supplies the “what,” and structure supplies the “where.” This is confirmed by our new graph-only experiments as follows, where structure alone fails on generative QA despite being useful in structure classification (§3.6).
> > >
> > >
> > > | Tuning | StructQA | Permute | Robustness |
> > > |---|---|---|---|
> > > | PT on graph | 7.00  | 7.13  | 14.60  |
> > > | PT on text | 37.8 | 29.93 | 31.07 |
> > > | PT on TAMO | 59.07 | 43.47 | 43.80 |
> > > | LoRA on graph | 7.93  | 7.00  | 14.80  |
> > > | LoRA on text | 45.93  | 35.87  | 39.67  |
> > > | LoRA on TAMO | 67.67  | 42.77  | 53.73  |
> > > | SFT on graph | 6.93  | 6.33  | 16.53  |
> > > | SFT on text | 62.00  | 42.86  | 51.53  |
> > > | SFT on TAMO | 69.40  | 43.74  | 64.07 |
> > >
> > > ---
> > >
> > > **2. Q2 – On comparison with DocLLM and similar models:**
> > >
> > > Layout-aware models like DocLLM focus on **visual document understanding**, processing OCR-extracted (text, bounding box) sequences to capture layout from images. This approach can be token-intensive for dense tables, as it requires encoding the 2D pixel coordinates for every cell. TAMO operates in a more foundational domain — **structured data reasoning** — where the table is already cleanly extracted, and the challenge is to capture logical structure that persists beyond any visual layout.
> > >
> > > These two approaches are complementary: *the output of a DocLLM-like system could serve as TAMO’s input.* TAMO's unique advantage lies in its ability to model the deep, logical relationships—like permutation invariance and hierarchy—that are independent of any visual layout. This is a problem that layout-aware models do not target.
> > >
> > > ---
> > >
> > > We hope this clarifies both the distinct roles of text and structure in TAMO, and how our approach complements, rather than replaces, other modalities.

---

> > > > ### Comment · Reviewer_v8d6 · 2025-08-09
> > > >
> > > > Thanks for solving my concerns and I hope this work is thoughtful for the community. I will raise my rating to 4

---

### Official Review · Reviewer_TiPM · 2025-07-07

**Clarity:** 3
**Significance:** 2
**Originality:** 2
**Rating:** 4
**Confidence:** 4

**Summary:**

This paper introduces TaMo, a framework that treats tables as a distinct modality to improve LLM performance on table reasoning tasks. Instead of flattening tables into text, TaMo uses a hypergraph encoder to capture structural relationships in the table. TaMo feeds these structure embeddings into the language model as additional input, improving its ability to reason over tables. Experiments across five benchmarks show that TaMo improves accuracy compared to standard LLM approaches.

The authors also introduce StructQA, a new benchmark to evaluate if permutation invariance is respected

**Questions:**

My main concern is the evaluation.
- The term “Specialist SoTA” is used in Table 2, but the paper does not discuss which specific models were used for each dataset. Please provide a breakdown of which models are used for with brief descriptions of how they were trained or evaluated.
- TaMo underperforms GPT-4 on WikiTQ, but this is not adequately discussed or analyzed - please include some minor comment on this
-  The paper does not include comparisons to well-established structure-aware baselines such as TAPAS/TAPEX. TAPAS and TAPEX are more relevant baselines than GPT-3.5 for table QA tasks
- The paper lacks ablations that show the contributions of TaMo’s components. Consider adding or discussing ablation experiments, this would clarify which parts of TaMo are important
- While generally clear, some aspects (e.g., multiset functions and alignment details) could be simplified or clarified further - please do that
- I would also like the authors to make more clear their contribution compared to prior works because it looks like the paper borrows a lot from existing approaches (e.g. hytrel)

**Ethical Concerns:**

["NO or VERY MINOR ethics concerns only"]

**Final Justification:**

The authors addressed most of my concerns and said they will update their manuscript accordingly.

They conducted additional experiments, included explanations/clarifications, and elaborated on previously vague statements.

I updated my score to reflect this.

However, I still have some concerns regarding originality/significance of the work.

**Limitations:**

The authors have addressed several limitations and I have no concerns regarding negative societal impact.

**Paper Formatting Concerns:**

The related work is at the end of the paper.

**Quality:**

3

**Strengths And Weaknesses:**

**Strengths**
- Proposes treating tables as a separate modality, enhancing structural reasoning in LLMs.
- Leverages hypergraph-based encoders to model hierarchy and permutation invariance — key properties of tabular data
- The authors show large performance gains overall
- Introduces StructQA, a new benchmark for testing if permutation invariance in tables is respected
- Plug-and-play integration with LLMs
- The paper is well-written and easy to follow

**Weaknesses**
- While the combination of parts is novel, many components (e.g., hypergraphs, structural encodings) are close extensions of existing work
- While generally clear, some aspects (e.g., multiset functions and alignment details) could be simplified or clarified further
- The evaluation does not seem thorough, the authors do not compare against other models that also incorporate table structure (e,g, TAPAS, TAPEX and others) and are baselines for table QA
- Specialist SoTA scores are included but not clearly explained, making it hard to assess the fairness of the comparison
- TaMo underperforms GPT-4 on WikiTQ, but this is not adequately discussed or analyzed
- Lacks ablation studies to isolate the contribution of key architectural components

---

> ### Author Rebuttal · Authors · 2025-07-31
>
> We sincerely thank the reviewer for their valuable time and constructive feedback. We appreciate that the reviewer found our paper well-written and easy to follow. We address the concerns below.
>
> **1. W4 / Q1 (Clarification of "Specialist SOTA"):**
>
> This is an excellent point. The term is indeed vague. In our revision, we will update the caption of Table 2 to explicitly name the specific model and provide its citation for each dataset. The specific models are:
>
> - For **HiTab**, the Specialist SOTA is **TableLlama**, which is based on Llama2-7B and underwent extensive supervised fine-tuning (SFT) on the HiTab training set along with other table QA datasets. The score reported in the original paper is 64.71, which we reproduced as 63.76.
>
> - For **WikiTQ** and **FetaQA**, the SOTA is **CABINET**, which improves LLM performance by using an external scorer to assign higher weights to question-relevant table cells, thereby filtering out noisy ones.
>
> - For **WikiSQL**, the SOTA is **SeaD**, which enhances text-to-SQL capabilities through a schema-aware denoising objective.
>
> It is important to note that our purpose in including these specialist SOTAs is to highlight different methodological directions. Unlike our TAMO framework, which directly enhances the LLM's end-to-end understanding of the table as a modality, methods like CABINET and SeaD rely on external modules or task-specific pre-processing.
>
> **2. W5 / Q2 (Performance on WikiTQ vs. GPT-4):**
>
> We thank the reviewer for this sharp observation. We will add a discussion of this point to Section 3.3. Our analysis is twofold. First, the WikiTQ dataset is sourced from early Wikipedia web pages, and its questions are often of a general nature. This creates a potential for data contamination, where a model as extensively trained as GPT-4 may have already encountered these tables or facts during its pre-training, allowing it to answer some questions without fully relying on the provided table.
>
> Second, as we initially noted, the performance gap is also attributable to the immense scale and powerful general reasoning capabilities of the proprietary GPT-4 model. However, our key finding remains valid: TAMO provides a substantial boost over its base model (Llama2-7B SFT: 45.81 vs. Text-only SFT: 31.22), proving the effectiveness of our approach. Furthermore, as shown in Appendix C.6 (Table 7), when TAMO is applied to a more capable base model like LLaMA 3.1 8B, its performance on WikiTQ improves significantly, demonstrating that TAMO's benefits scale with the underlying LLM's power.
>
> **3. On Comparisons with TAPAS/TAPEX (W3, Q3):**
>
> We agree that comparing with seminal structure-aware models like TAPAS and TAPEX is important. Our initial focus was on the modern **decoder-only LLM-based paradigm**, thus we prioritized comparisons with methods that also use serialized inputs (e.g., GPT-4, TableLlama). To further strengthen our evaluation, we have conducted additional experiments comparing TAMO with TAPAS and TAPEX on our StructQA benchmark. The results are as follows:
>
> | Model | StructQA (Accuracy) | Permute (Accuracy) | Robustness |
> |---|---|---|---|
> | **TAPAS** | 4.67 | 4.67 | 65.47 |
> | **TAPEX** | 12.33 | 9.6 | 54.73 |
> | TAMO | 59.07 | 43.47 | 43.8 |
>
> The results clearly show that TAMO dramatically outperforms these baselines on the core structural understanding tasks (StructQA and Permutation Accuracy). It is crucial to consider the architectural differences: TAPAS is a BERT-based encoder, and TAPEX is a BART-based encoder-decoder model. In contrast, our TAMO framework is specifically designed as a multimodal extension for **decoder-only LLMs**, which represent the current dominant architecture. While TAPAS and TAPEX show higher robustness, this is largely an artifact of their very low base accuracy; it is easier to remain consistent when a model is consistently incorrect. TAMO's performance demonstrates a far superior grasp of table structure. We will add a new table and this analysis to the appendix in our revised manuscript.
>
> **4. On Ablation Studies (W6, Q4):**
>
> We would like to respectfully clarify that our main experimental design in **Table 2** constitutes a core ablation study. Across four different settings (Inference-only, Prompt Tuning, LoRA, SFT), we directly compare the performance of using text-only inputs versus our TAMO approach, which adds the table structure modality. The consistent and significant gains demonstrated across all settings (e.g., +21.27 absolute accuracy on StructQA for Prompt Tuning vs. TAMO) serve as the primary evidence for our method's contribution. We also provide more fine-grained analyses in **Section 3.6 (encoder effectiveness)** and **Appendix C.3 (impact of token numbers)**.
>
> **5. On Novelty vs. HyTrel and Clarity (W2, Q5, Q6):**
>
> We appreciate the request for clarification. We respectfully point the reviewer to Appendix D.1, "Positioning of TAMO", where we provide a detailed discussion and a summary table (Table 8) to articulate the fundamental difference. In short, HyTrel is a modality encoder for learning table representations, whereas TAMO is a complete multimodal reasoning framework that integrates such an encoder into an LLM to solve joint text-table tasks (e.g., QA). This is a non-trivial conceptual leap, analogous to the distinction between ViT (an encoder) and LLaVA (a multimodal LLM). To ensure this crucial point is not missed, we will add a forward-reference to Appendix D.1 in our main methodology section. We will also revise Section 2.2 to further clarify the mathematical notations to improve readability.

---

> > ### Author Response · Authors · 2025-08-06
> > **Follow-up by Authors**
> >
> > Dear Reviewer,
> >
> > We are writing to follow up on our rebuttal and hope that we have comprehensively addressed your initial questions and concerns.
> >
> > Several rounds of public discussion with other reviewers have helped clarify our contributions and methodology further; we believe they may also address potential follow-up concerns from your side.
> >
> > In short, our work introduces a new perspective to the TableQA and LLM communities. Beyond the standard practice of representing tables via text serialization, we propose using table embeddings to provide the LLM with more comprehensive and robust structural understanding. We hope this initial exploration can inspire the community to think beyond the limitations of text-only table representations.
> >
> > We truly value this discussion process and are committed to incorporating insights from all reviewer conversations into the final version. We believe this will greatly help readers better understand our work's context, strengths, and limitations.
> >
> > We hope our clarifications have been satisfactory. If you have any further questions, please do not hesitate to let us know—we would be happy to discuss them.

---

> > > ### Comment · Reviewer_TiPM · 2025-08-09
> > >
> > > Thank you for your answers. I will adjust my score.

---

### Note · Authors · 2025-08-14

We sincerely thank the reviewers for their invaluable feedback. We are grateful that our detailed rebuttal and new experiments have successfully **resolved the main concerns about experimental comparisons, contribution clarity, and ablation studies**.

The core discussion centered on our contribution. We argue our novelty lies in a cohesive scientific narrative: (1) We **diagnosed a critical limitation**, showing with new experiments that structural information loss from text serialization is a fundamental issue persisting across classic (e.g., TAPAS, TAPEX) and SOTA models (e.g., GPT-4.1). (2) We proposed TAMO, a **principled multimodal solution** that treats tables as a distinct modality, feeding embeddings from a dedicated structural encoder into the LLM. (3) Our new ablations yielded a **pivotal insight**: this synergistic design is necessary, as text-only approaches fail on structural invariance while graph-only methods lack semantic context for generative QA. Ultimately, our work highlights the untapped potential of table embeddings for comprehensive LLM understanding and offers effective insights for future research.

According to the NeurIPS reviewer’s guideline, “*Originality does not necessarily require introducing an entirely new method. Rather, a work that provides novel insights by evaluating existing methods, or demonstrates improved efficiency, fairness, etc. is also equally valuable.*” We believe our work meets this standard by systematically studying how LLMs can better understand tables. Encouragingly, reviewers found our clarifications and new experiments **effective and insightful for the community**, leading all five to confirm their concerns were resolved and that they would adjust their scores positively.

- Reviewer 7uvU confirmed our response "answers my question well".
- Reviewer v8d6 called our work "thoughtful for the community" and would "raise my rating to 4."
- Reviewer Qmvd found our results "encouraging and very positive".
- Reviewers vZ6u and TiPM all confirmed they would adjust their scores appropriately.

This productive discussion has greatly improved our paper, and we are fully committed to incorporating all new experiments and clarifications into the final version. We kindly ask the Area Chair to consider the strong positive consensus from the reviewers, and we believe our work makes a strong and practical contribution to the field.

Thanks for the Area Chairs' and all reviewers' careful consideration.

---

### Decision · Program_Chairs · 2025-09-17

**Decision:**

Accept (poster)

**Comment:**

The authors study an interesting idea: rather serialize table as text input as existing works, they adapt a multimodal approach, by treating as a different modality. The reviewers found the paper novel, and the experiments are well conducted.